# Frailty in Geriatrics: A Critical Review with Content Analysis of Instruments, Overlapping Constructs, and Challenges in Diagnosis and Prognostic Precision

**DOI:** 10.3390/jcm14061808

**Published:** 2025-03-07

**Authors:** José Fierro-Marrero, Álvaro Reina-Varona, Alba Paris-Alemany, Roy La Touche

**Affiliations:** 1Department of Physiotherapy, Centro Superior de Estudios Universitarios La Salle, Universidad Autónoma de Madrid, 28023 Madrid, Spain; jose.fierromarrero@yahoo.com (J.F.-M.); alvaroreina93@gmail.com (Á.R.-V.); roylatouche@yahoo.es (R.L.T.); 2Motion in Brains Research Group, Centro Superior de Estudios Universitarios La Salle, Universidad Autónoma de Madrid, 28023 Madrid, Spain; 3PhD Program in Medicine and Surgery, Doctoral School, Universidad Autónoma de Madrid, 28029 Madrid, Spain; 4Department of Radiology, Rehabilitation and Physiotherapy, Faculty of Nursery, Physiotherapy and Podiatry, Complutense University of Madrid, 28040 Madrid, Spain; 5Instituto de Dolor Craneofacial y Neuromusculoesquelético (INDCRAN), 28008 Madrid, Spain

**Keywords:** frailty, geriatrics, disability, morbidity, instrument development, construct assessment, content validity

## Abstract

Frailty is a key concept in geriatric care; yet its definition and assessment remain debated. Since the early 2000s, two main models have emerged: the Fried frailty phenotype, focusing on physical deficits, and the Mitnitski frailty index, which incorporates broader health factors. These divergent approaches have led to over 50 frailty instruments, reflecting the absence of a unified framework. This review explores the content, weighting, and scoring methods of frailty instruments, identifying potential concerns derived from this. This review exposes the overlap of frailty with other constructs including function, disability, morbidity, and sarcopenia. Many instruments lack content validity, and detect highly heterogeneous samples within and between scales, all labeled under the “frail” tag. This poses challenges to interpreting instrument responsiveness. In addition, frailty should not be considered a clinical entity with a unique etiology. This review discusses how the broad nature of frailty conflicts with modern paradigms of individualization and precision. They may be useful in primary care, but lack the specificity for secondary care evaluations. This article also discusses how the predictive validity of frailty should be interpreted with caution. Finally, we summarize our findings and propose a new definition of frailty, highlighting the strengths and weaknesses of the construct. The identified inconsistencies should serve as a guide for refining the concept of frailty, both in research and in its application to geriatric care.

## 1. Introduction

The term “frailty” in the context of aging has been a subject of significant interest and debate within the field of geriatrics. Its elusive nature has intrigued scholars, clinicians, and scientists, inciting them to more deeply understand, discuss, and define this multifaceted construct. What originally appeared to be a relatively straightforward term has quickly evolved into a complex construct characterized by diverse interpretations with a remarkable absence of consensus.

The concept of frailty began its rise in the 2000s, when two separate research groups independently popularized the term in the medical lexicon [1]. Although this pioneering research alluded to the term “frailty”, they presented distinctly different viewpoints. This early divergence in conceptualization set the stage for a debate that continues to this day.

Over time, numerous conceptual and operational definitions of frailty have emerged, reflecting the evolving nature of this field. Researchers have approached frailty from diverse perspectives, ranging from physical to multidimensional frameworks. Consequently, the essence of frailty has remained elusive, with the scientific community struggling to achieve a unified understanding.

Efforts have been made to integrate the various definitions, most notably through Delphi consensus studies, which have aimed to determine the commonly agreed conceptual spheres of frailty. Despite these efforts, a comprehensive theoretical definition for frailty remains unclear. This gap in the understanding of frailty presents challenges to developing assessment tools and designing effective interventions and care strategies for frail older adults.

### Objective

The objective of this narrative review is to critically examine the historical evolution, varying definitions, and assessment tools of frailty, highlighting the significant inconsistencies in its conceptualization and operationalization. We aim to identify the procedures concerning the content validity of instruments, possible inconsistencies in sampling profiles, scale responsiveness; examine the overlap of frailty with related constructs such as disability, morbidity and sarcopenia; and explore the predictive validity of frailty with health outcomes. With this, we will discuss if frailty can be considered a clinical entity, its etiological basis and if it suits modern medicine paradigms. Finally, we will provide a summary of the findings with the conclusion of a new proposed definition of frailty based on its strengths and limitations. By critically analyzing the construct of frailty, this review seeks to guide future research in developing useful assessment tools, ultimately improving the predictive validity and clinical utility of frailty in geriatric care.

## 2. Methods

### 2.1. Search Strategies

Searches were conducted in PubMed and Google Scholar in November 2023 with the following aims: (1) to identify potential studies exploring the definitions of frailty, mainly through review designs; (2) to identify which current instruments are available in the literature to assess frailty; and (3) to search for the prognostic association of frailty with health-related adverse events.

### 2.2. Information Synthesis

The data identified through the search process were carefully screened, extracted, and organized into a comprehensive format. Information was synthesized narratively and visually, employing figures and tables for clarity. Definitions of frailty were systematically gathered and analyzed, focusing on their conceptual frameworks and operationalization.

In order to explore the consistency within and between frailty assessment tools, we explored the content, assessment procedures, weighting and scoring systems of frailty scales. This process was conducted analyzing the scales identified by Faller et al. [2]. The content of items were analyzed employing a bottom-up approach, clustering into conceptual spheres. Furthermore, comparisons were made to explore similarities and discrepancies across different instruments. The associations between frailty and health-related adverse outcomes, such as morbidity, disability, and mortality, were critically examined and presented in tabular format to facilitate interpretation. This process enabled a structured narrative analysis of the key findings, highlighting trends, gaps, and areas for future research.

## 3. Results

### 3.1. Historical Footsteps of Frailty

Numerous studies have explored the historical origins of the concept of frailty [3,4,5], pointing to the last quarter of the 20th century as when this concept was originally conceived. However, it was not until 2001 that two distinctive conceptualizations of frailty emerged. Mitnitski et al. [6] developed a frailty index based on the accumulation of health-related deficits, whereas Fried et al. [1] proposed a frailty phenotype based on the presence of a specific set of signs, symptoms, and findings.

The frailty index assessed frailty as a cumulative burden of multiple health-related deficits, including physical functioning, cognitive performance, emotional well-being, participation in activities of daily living (ADL), sensory perception, comorbidities, and plasmatic marker assessments. This approach conceptualizes frailty as a continuous variable, with a higher number of accumulated deficits indicating greater vulnerabilities. In contrast, the frailty phenotype identifies frailty through the presence of at least three of five key findings: unintentional weight loss, exhaustion/fatigue, low physical activity level, slow gait speed, and low hand-grip strength. Unlike the frailty index, which provides a gradual risk stratification, the frailty phenotype classifies individuals into three discrete categories: frail (≥3 findings), pre-frail (1–2 findings), and robust (0 findings).

The introduction of these two models marked a turning point in the literature, leading to the conceptual distinction between physical frailty, characterized by assessing physical components, and multidimensional frailty, which encompasses deficits across multiple health domains.

These were the first attempts to operationalize the concept of frailty through assessment tools, initiating a continuous process of development and refinement. As a result, there are now 51 different measurement tools to assess frailty in older adults [2]. In clinical populations, frailty has gained attention as a critical determinant of health outcomes, influencing the prognosis and the management of patients. As a result, classical assessment tools from geriatrics have been adapted, and new condition-specific scales have been developed to assess frailty. For example, frailty tools have been specifically designed for hip fracture [7], heart failure [8], breast cancer [9] or stroke [10] among others.

The extensive number of available tools highlights the lack of consensus on what frailty entails, a concern repeatedly highlighted in the literature by authors such as Rockwood et al. [11], Vass and Hendriksen [12], and Apostolo et al. [13].

In recent years, new concepts have emerged in geriatric care. Terms such as “physiologic reserve”, “intrinsic capacity”, “physical resilience”, and “functional ability” are key factors that influence health and independence in older adults [14,15]. While these concepts offer a valuable framework for geriatrics, their measurement and quantification in a standardized meaningful way remain difficult.

### 3.2. The Construct of Frailty

It should be noted that frailty, just as any other construct, can be encompassed within theoretical, conceptual, and operational frameworks, which typically have associated definitions.

Theoretical definitions address the abstract and foundational explanations of the construct. Although these definitions might lack precision, they are useful for explaining the essence, nature, and principles of the construct [16].

Conceptual definitions provide a more precise and detailed description of the construct’s key components, or conceptual spheres. They are typically established within a specific discipline or context. These definitions are useful for researchers and scholars to ensure a common understanding of the concepts surrounding the construct. Additionally, conceptual definitions bridge the gap between theoretical and operational definitions [16].

Operational definitions describe the procedure to materialize, measure, and manipulate the original construct, making it suitable for experimental research. They provide the methodology and criteria to measure the original construct. Operational definitions are based on conceptual definitions, which in turn are based on theoretical definitions (see Figure 1A).

To highlight these differences, consider the concept of “happiness.” A theoretical definition of happiness might discuss its philosophical, religious, cultural, and psychological aspects. On the other hand, a conceptual definition could break down happiness into its various components, such as emotional well-being, life satisfaction, and fulfilment, describing how these spheres interrelate. Different authors might offer various ways for conceptually understanding happiness, proposing different spheres. An operational definition puts these definitions into practice, for example, by measuring happiness with the Short Happiness Questionnaire on a 0–100-point basis (see Figure 1B).

#### 3.2.1. Theoretical, Conceptual, and Operational Definitions of Frailty

Numerous authors have proposed their interpretations of frailty in the literature. Due to the diversity of viewpoints on the essence of frailty, several expert panels have convened to develop consensus-driven definitions. Examples of such are the expert panel organized by Morley et al. [17] and the Delphi study conducted by Rodríguez-Mañas et al. [18]. Additionally, in 2019, the International Conference on Sarcopenia and Frailty Research assembled another panel of experts culminating in the proposal of a new definition of physical frailty [19]. It is worth noting that all these expert panels reached agreement solely on the conceptual and operational definitions of frailty.

Several systematic reviews have analyzed frailty definitions [4,20,21,22]. However, only two of them [4,21] explicitly extracted and compiled all identified frailty definitions within their reports. The definitions analyzed in the four reviews were primarily conceptual and operational. Although one review claimed to have identified theoretical definitions of frailty, a closer examination of its “Theoretical definition” section reveals that the synthesized definitions were, in fact, conceptual [22].

The lack of a consistent theoretical definition of frailty within the scientific literature has significantly influenced the trajectory of our investigations. This lack of a solid theoretical foundation has led us down a misguided path, fostering a false sense of understanding regarding the true meaning of the term. As previously mentioned, this eagerness has resulted in the development of a wide range of frailty evaluation tools [2] and expert panels reaching consensus on the assessment procedures that these tools should include [23]. Although this suggests an agreed-upon operational framework, it still lacks a coherent theoretical definition.

Although no theoretical definition of frailty has been established, several systematic reviews have qualitatively analyzed its conceptual domains based on the available definitions [20,21,22], reaching similar conclusions. Another interesting definition with several concepts is the one proposed by the World Health Organization (WHO), which states:

“*Frailty may be conceptually defined as a clinically recognizable state in older people who have increased vulnerability, resulting from age-associated declines in physiological reserve and function across multiple organ systems, such that the ability to cope with everyday or acute stressors is compromised*”.[15]

This definition highlights key concepts such as clinical state, vulnerability, and decline. Figure 2 provides a visual representation of the concepts surrounding frailty.

#### 3.2.2. Physical vs. Multidimensional Frailty

Over the years, the term physical frailty has become widely used to describe the concept initially proposed by Fried et al. [1] through the “frailty phenotype” model [17,19,24,25,26]. We currently have various assessment tools designed to identify this physical frailty [2], also known as the Cumulative-Deficit model, introduced by Mitnitski et al. [6]. This alternative model was later conceptualized as multidimensional frailty, and multiple assessment tools have since been created for this approach as well [2].

These two approaches to frailty differ fundamentally in their conceptual framework. The conceptual spheres surrounding physical frailty focus on physical health-related deficits, whereas multidimensional frailty encompasses not only physical deficits but also cognitive, emotional, social, and spiritual deficits [18].

It is important to note that these concepts are not entirely clear and well defined. Significant controversies exist surrounding the operationalization of physical frailty assessment tools due to the lack of a unified theoretical framework. For instance, the canonical frailty phenotype of Fried et al. [1] cannot be strictly ascribed within the physical frailty model alone. This instrument includes five items, of which only four could be considered physical assessments, specifically targeting physical structures, physical functions, and activity, based on the categorization proposed by the International Classification of Functioning, Disability and Health (ICF) [27].

Physical structures’ evaluation: weight loss.Physical functions’ evaluation: hand grip strength and gait speed.Activity evaluation: physical activity level.

The “exhaustion” (fatigue) item cannot be fully categorized as a purely physical assessment domain. Borg et al. [28] suggested long ago that fatigue is a multidimensional construct influenced by various factors, including muscle fatigue, cardio-respiratory exertion, and psychological distress, see Figure 3.

A more in-depth analysis of controversies within and between physical frailty and multidimensional frailty assessment tools will be addressed in the following sections.

### 3.3. The Validity and Limitations of Frailty Instruments

The Consensus-based Standards for the Selection of Health Measurement Instruments (COSMIN) panel defines content validity as “*the degree to which the content of a measurement instrument is an adequate reflection of the construct to be measured*” [29]. Initially, COSMIN emphasizes what they call “face validity,” which refers to “*the degree to which the items of a self-reported outcome instrument properly reflect the construct themselves*.” This qualitative assessment should give the reader the impression of the construct being assessed by only reading its items. After passing this face validity check, a panel of experts should evaluate the comprehensiveness, coherence, measurement procedure, and relevance, among other domains, of the items included in the instrument to assess the construct [30].

The COSMIN authors emphasize that a proper content validity process requires a thorough description and definition of the construct [30]. This has been a lacking constant in the history of the frailty construct, affecting the development of assessment tools. The systematic review by Faller et al. [2] aimed to identify and gather all tools for detecting frailty in older adults. Among the 51 frailty assessment tools identified, only 15 included a content validity process during their development. We hypothesize that the absence of this process may rely on the assumption that frailty was universally understood, and there was no need to proceed with a content validity process.

It is probable that popularity of a few frailty tools, such as the frailty phenotype [1], created a snowball effect, leading to the development of new assessment tools based on the original instrument without further content validity processes [2,31]. Readers should be aware that popular frailty instruments, such as the frailty phenotype [1], the frailty index [6], and the Clinical Frailty Scale [32] did not undergo any content validity processes [2]. This lack of rigor in the development of frailty assessment tools is probably one factor contributing to the heterogeneity between and within instrument items, blurring the essence of what should be considered frailty.

#### 3.3.1. Inconsistencies in Instruments’ Content

Of the 51 instruments identified by Faller et al. [2], we were able to analyze the content of 50. Additionally, we examined two shortened versions of the missing instrument. Based on this analysis, the items within the frailty instruments were categorized into distinct groups, as shown and described in Figure 4. This classification was conducted using evaluation categories established by three major frameworks: the ICF [27], the International Classification of Diseases 11th Edition (ICD-11) [33], and the Occupational Therapy Practice Framework: Domain and Process 4th Edition by the American Occupational Therapy Association (AOTA-4) [34].

As observed, the range of assessments is extensive, encompassing several domains. These include anthropometric measures, a wide array of body functions (such as neuromotor, respiratory, or visual functions), diseases, symptoms and sighs (such as neoplasms, or diseases of the respiratory or the nervous system among others), assessments of social participation, educational, or rest and sleep features, along with limitations or dependency across ADL, IADL, and health management. Finally, environmental or personal factors (lifestyle, socioeconomic status, gender or ethnicity, among others) were also identified, along with other non-classifiable factors.

Based on the analysis presented in Figure 4, a broader classification of items was conducted to provide a more comprehensive assessment of instruments’ content. We refined classification categories of items into the following groups: body structures; body functions; diseases, symptoms and signs; social participation; education; rest and sleep; limitations or dependency in ADL; limitations or dependency in IADL; limitations or dependency in health management; environmental factors; personal factors; and others. This new classification is presented in Figure 5.

It is important to consider which types of assessments are included in a scale. As observed in Figure 4 and Figure 5, there is a significant heterogeneity across items within and between instruments. For instance, many frailty instruments assessed body structures: mainly weight loss, body mass index, and waist circumference. Body functions, such as neuromotor functions, cognitive functions, emotions and motivation, or plasmatic state are assessed in most instruments. The history of disease is a common assessment in a body of instruments, and is the most predominant assessment in the electronic frailty index, Frailty Index-Canadian Study of Health and Aging, and Care Partner-Frailty Index-Comprehensive Geriatric Assessment [35,36,37]. Social participation, education, rest, and sleep were the least prevalent assessments. Limitations or dependency in ADL and Instrumental Activities of Daily Living (IADL) were also prevalent among frailty scales [38,39,40,41]. Other scales, such as the Comprehensive Frailty Assessment Instrument, presented a great part of their evaluations addressed to environmental factors [42], while the Tilburg Frailty Indicator presented broader evaluations of personal factors [43].

#### 3.3.2. Inconsistencies in Assessment Procedures, Scoring Systems and Weightings

Another important aspect to address is the variation in procedures to assess specific variables, the scoring method, and the weighting of items within frailty scales. To expose these inconsistencies, we focused on a commonly assessed variable in frailty instruments: weight loss.

After a deeper review of the published scales, we found that the majority evaluated this variable through self-reports (Edmonton Frailty Scale, Frail Non-Disabled instrument, 5-item FRAIL scale, Mini-Nutritional Assessment, Groningen Frailty Indicator, Frailty Index for Elders) [44,45,46,47,48,49], whereas others rely on qualitative interviews (Clinical Global Impression of Change in Physical Frailty) [50]. The assessment procedure is important, given that it can bias the collected data. For example, self-reported methods to assess anthropometric variables, such as height or weight, become inaccurate with aging (>60 years) [51,52]. Moreover, research suggests that self-report bias can stem from cognitive burden and recall bias, especially compared with more objective data collection methods [53], a concern that is further pronounced in cognitively impaired individuals.

The scoring of items such as weight loss also varies across instruments. For example, although weight loss is a continuous variable, all instruments categorize the response options, which limits the precision of the assessment. Most scales use dichotomous (yes/no) response options (Electronic Frailty Index, Continuous Frailty Scale, Edmonton Frailty Scale, Frail Non-Disabled instrument, 5-item FRAIL scale, Groningen Frailty Indicator, Kaigo-Yoobo Checklist, and Frailty Index for Elders) [32,35,44,46,47,48,49,54], whereas others include additional response categories (Care Partner-Frailty Index-Comprehensive Geriatric Assessment, and Mini-Nutritional Assessment) [36,45] or rely on broader qualitative examinations by clinicians without predefined scoring categories (Clinical Global Impression of Change in Physical Frailty) [50]. Previous literature has suggested that, despite the potential bias in weight assessment through self-reports, if used, the authors recommend that weight loss should be analyzed as continuous data rather than being categorized [52].

Additionally, the weighting of items is relevant for the total scale score, because it can assign different levels of importance to various items. Weight loss is given equal weight to other items in scales, such as the Electronic Frailty Index, Continuous Frailty Scale, Edmonton FRAIL scale, Frail Non-Disabled instrument, and 5-item FRAIL scale [32,35,44,47,48]. In contrast, it is weighted unequally in the MNA Mini-Nutritional Assessment [45], or subjectively weighted by the evaluator for the final scoring for the Clinical Global Impression of Change in Physical Frailty [50].

#### 3.3.3. Heterogeneity in Sampling Profiles and in Scale Responsiveness

The variability in the content of the scales, along with the weighting and scoring method across all instruments, results in significant heterogeneity in the samples identified both within and between scales.

Within a single scale, patients can be classified as frail based on different criteria. This means that patients with distinct characteristics may all meet the criteria for frailty, even if the underlying factors are different. For example, one patient might be considered frail due to deficits in disability-related items, while another might be identified as frail based on anthropometric measurements. These patient profiles are fundamentally different; however, the same “frailty” label is applied to all of them. This pose concerns whether frailty patients within a single or different scales are comparable.

In fact, the variability is even more pronounced between different scales, as discussed in previous works [55]. Each instrument employs unique criteria (Figure 4 and Figure 5), leading to differences in the patients identified. For example, a patient might be classified as frail when using the frailty phenotype but not when using the frailty index, potentially leading to different clinical decisions. This has been evidenced in previous epidemiological studies, as the prevalence of frailty highly varies depending on the employed instrument. For example, the prevalence of frailty participants in Norway varies from 35.8% to 10.6% when using the 35-item frailty index or the frailty phenotype, respectively [56]. In addition, it not only varies between physical and multidimensional frailty instruments, but additionally within them, as the prevalence varies from 18.7% to 27.8% by employing the 5-item FRAIL scale, or the frailty phenotype, respectively [57]. Figure 6A summarizes how different profiles are detected within and across scales.

Another issue arises from the heterogeneity of samples identified depending on the scores within a scale. Samples with lower scores will exhibit greater heterogeneity among themselves, while those with higher scores will tend to be more homogeneous, as they will share more traits. Most frailty scales employ cut-off scores to distinguish between prefrail (lower scores) and frail (higher scores) categories. As a result, prefrail samples, both within and between scales, are more heterogeneous than frail groups. See Figure 6B.

A final issue concerns the interpretation of changes in an individual or a sample of subjects when the score varies over time, also known as responsiveness. Unlike unidimensional scales, most frailty instruments assess multiple dimensions (as suggested by their content). A change in a unidimensional scale indicates a change in the intensity or frequency of the single evaluated trait. In contrast, a change in a frailty scale does not clarify which specific dimension is being modified. Therefore, several interpretations should be considered:**Change in the total score of frailty:**○**Change in a single dimension:** The variation in the score might reflect a change in one specific dimension assessed by the scale, such as morbidity, disability or physical functioning. This change could occur differently across individuals within a sample.○**Change across multiple dimensions:** A change in the total score could result from simultaneous variations in multiple dimensions. These dimensions might all change in the same direction (e.g., all improving or worsening), or there could be a mix where some dimensions improve while others worsen or remain unchanged. This counterbalancing effect still leads to an overall change in the total score. Such combination of variations can differ significantly across individuals within a sample.**No change in the total score of frailty:**○**No change in any dimension:** The total score may remain unchanged because none of the assessed dimensions have varied. In this case, the stability of the total score reflects a lack of change across all measured domains.○**Counterbalancing changes across dimensions:** The total score could also remain stable despite significant variations within dimensions. For instance, some dimensions may improve, others may worsen, and some may remain unchanged. These opposing changes offset each other, resulting in no observable change in the total score, even though meaningful changes have occurred in the underlying dimensions. These variations may differ between individuals in a sample.

The exposed concerns reveal several problems for frailty scales, as they are detecting highly heterogeneous profiles, and their responsiveness could not be precise enough for proper interpretation of results (Figure 7).

### 3.4. Frailty Overlaps with Many Other Constructs

As previously mentioned, frailty is based on the assessment of specific health deficits, often incorporating elements from other constructs. The most frequently included constructs are function, disability and morbidity, which have been widely discussed in the literature [58]. For example, limitations or dependency in ADL and IADL are commonly assessed in frailty scales, yet these are also central to disability assessments, even with authors stating that frailty could exclusively encompass limitations in activities (in other words, disability) [59,60].

However, frailty does not only overlap with these three constructs but also with many others, as illustrated in Figure 5. Therefore, we will examine how frailty overlaps with these constructs and explore scenarios where frailty may be distinguished as a unique concept.

#### 3.4.1. Frailty vs. Disability and Functioning

The term “disability” has undergone a rich historical evolution, with authors addressing it from religious, folkloric, medical, political, and social activism perspectives [61]. For a more in-depth discussion, please refer to Riddle [62].

To fully comprehend what disability entails, we should review the various definitions proposed throughout history. The WHO defined disability in the International Classification of Impairments, Disabilities and Handicaps (ICIDH) as follows:

“A disability is any restriction or lack (resulting from an impairment) of ability to perform an activity in the manner or within the range considered normal for a human being” [63].

The ICIDH, in further conceptualizing the term “disability,” described it as a restriction or lack in the following: “compound or integrated activities expected of the person or of the body as a whole, such as are represented by tasks, skills, and behaviors.” They further distinguished disability from a focus on physical or physiological mechanisms: “disability represents a departure from the norm in terms of performance of the individual, as opposed to that of the organ or mechanism,” reiterating the emphasis on behavior and activities: “the concept is characterized by excesses or deficiencies of customarily expected behavior or activity”.

Since this definition, the focus of shifted toward an activity framework, moving away from the physical or physiological constructs. In 2001, the WHO updated this definition in the ICF, defining key terms:○Body Functions: Physiological functions of body systems, including psychological functions.○Body Structures: Anatomical parts of the body, such as organs, limbs, and their components.○Activity: The execution of a task or action by an individual.○Participation: Involvement in a life situation.○Environmental Factors: The physical, social, and attitudinal environment in which people live and conduct their lives.

Additionally, the following terms were defined:
○Impairments: Problems in body function or structure, such as a significant deviation or loss.○Activity Limitations: Difficulties an individual may have in executing activities.○Participation Restrictions: Problems an individual might experience from involvement in life situations.

Based on these concepts, the ICF defined “disability” as:

“*Dysfunctioning at one or more of the following levels: impairments, activity limitations, and participation restrictions*”.[27]

This new definition expanded the concept to include not only limitations in activity or participation but also any impairment in body function or body structures.

There have been numerous debates addressing whether frailty is conceptually the same as disability. Some authors suggest that frailty represents a limitation in practical or social activities, such as in ADL [59,60], suggesting that frailty and disability would represent the same concept. This prompts an important consideration: if frailty and disability were the same, using two terms to describe a single construct would be redundant and unnecessary. In such a scenario, the logical conclusion would imply the removal of one of those words, probably implying the removal of the word “frailty” due to its recent appearance, compared with the historically consolidated word, “disability”. Therefore, for the word “frailty” to stay afloat, its construct should offer a different perspective than just meaning “disability.”

This concern was addressed by other authors, who rejected frailty and disability as being the same. Several expert panels have indicated “*frailty and disability were not the same*” [58]; or stated “*frailty is not disability*” [17], “*frailty is different from disability*”, or “*frailty is a dynamic process, nonlinear, different from vulnerability and disability*” [18]. The Morley et al. [17] panel emphasized the following distinctions: “*frail individuals could be disabled*”, and “*not all disabled persons are frail*”.

#### 3.4.2. Frailty vs. Morbidity

The relationship between frailty and morbidity has been discussed recently, with some authors providing some insights into how they conceptually interrelate [64]. They have provided three conceptual frameworks as to how frailty and morbidity could be related. Our analysis of frailty instruments (Figure 4 and Figure 5) aligns with the proposal of Cesari et al. [64], given that many scales include the assessment of diseases within frailty measures such as the frailty phenotype, the 11-item FI, and the Prognostic Frailty Score, among others [47,65,66]. This inclusion suggests that frailty might be a broader construct integrating the evaluation of diseases. Conversely, other scales do not assess the presence of diseases, indicating that the constructs of frailty and morbidity could be considered independent.

#### 3.4.3. Frailty vs. Sarcopenia

Researchers have also attempted to establish the relationship between sarcopenia and frailty, with some considering sarcopenia as a cause of frailty [67]. Historically, sarcopenia has been defined by the loss of muscle mass, with some definitions also incorporating declines in strength and physical function. However, recent definitions have excluded the muscle mass as a diagnostic criterion due to the challenges associated with its assessment, instead focusing on muscle strength and functional declines as key indicators of sarcopenia [68]. It is important to note that, regardless of the specific sarcopenia criteria considered (whether muscle loss, strength loss or functional decline), these same factors are also included in certain frailty instruments making them shared evaluation components. For instance, the Fried phenotype includes assessments of hand-grip strength, and gait speed, both of which are also shared in sarcopenia assessments [68]. Therefore, while sarcopenia and frailty are distinct conditions, as stated with their respective definitions, they share overlapping assessments with certain frailty tools. Given these similarities, it is plausible that both conditions also share common underlying mechanisms, particularly with the Fried phenotype [69].

#### 3.4.4. Determining the Relationship Between Frailty and Other Constructs

Considering the lack of consensus in the literature regarding the relationship between disability, morbidity, and frailty, we present a diagram establishing an operational and conceptual framework based on the assessments included in frailty instruments (see Figure 8). Employing a bottom-up analysis focusing on the content of frailty scales, and the weighting and scoring systems, we can determine the relationship between constructs.

Although the diagram focuses on the relationship between disability and frailty, the same logic can apply to other constructs such as morbidity. Figure 8 illustrates various scenarios in which frailty is assessed in relation to disability items:**Frailty instrument excludes disability items:** In this scenario, frailty and disability are considered different constructs. Both constructs can be either interrelated or independent, meaning they may be present simultaneously or separately in a patient. Instruments such as the Frailty Trait Scale [70] or the Brief Frailty Index [71] do not include assessments of disability (considered as limitations in ADL or IADL); therefore in this case, frailty would be conceptually different from disability.**Frailty instrument exclusively comprises disability items:** Here, frailty and disability are operationally and conceptually the same. This redundancy implies that both constructs assess identical aspects, making them redundant. No identified instruments present this characteristic; however, some authors have conceptualized frailty being the same as disability. A definition by Raphael et al. [60] states “frailty is a diminished ability to carry out the important practical and social activities of daily living”.**Frailty instrument includes both disability and other items:** This is the most common method to assess frailty across the identified instruments. This scenario allows for three types of relationships based on the scoring systems and weightings within the scale:○**Disability as a component but not essential:** if the instrument includes disability items but does not require a positive score on these items for a frailty diagnosis, frailty is broader than disability and includes other health dimensions. In some patients, frailty could be explained exclusively by disability, exclusively by other factors, or by a summation of disability with other factors. All of the identified instruments present this type of weighting and scoring method.○**Disability as a necessary component:** if the instrument includes disability items and requires a positive score on these items for a frailty diagnosis. Frailty could be explained exclusively by disability, or by the simultaneous presence of disability with other factors. Among the gathered instruments, none presented this method.○**Disability and other factors as necessary components:** if the instrument includes disability and other items, with both being required to be positive, frailty will be explained by the mutual presence of disability with other factors. None of the identified instruments presented this method.

Multiple studies have analyzed if frailty can exist on its own. However, this construct can only be explained by the presence of deficits in other health-related constructs. Therefore, frailty will always merge from the presence of other underlying constructs.

### 3.5. The Predictive Validity of Frailty

#### 3.5.1. Assessing the Ability of Frailty to Predict Adverse Events

Despite the high degree of heterogeneity within and between frailty instruments, all exhibit prospective associations with various health-related adverse outcomes. These outcomes include mortality, hospitalization, institutionalization, disability in basic ADL (BADL), disability in IADL, falls, fractures, physical limitations, emergency department visits, dependency, cognitive decline, life satisfaction, and body composition [72,73]. Table 1 presents the results of Vermeiren et al. [72] and Yang et al. [73] regarding the prospective associations of frail, prefrail, and combined statuses in older adults (≥65 years) identified using grouped physical and multidimensional frailty assessment tools.

Examining more closely the prospective association of physical and multidimensional frailty tools, we can see that both types of tools are associated with mortality, hospitalization, institutionalization, disability in BADL, and disability in IADL [72,73]. However, meta-analyses indicate that physical and multidimensional frailty tools can show different prospective associations. For instance, physical frailty tools present positive associations with the risk of emergency department visits, falls, and fractures, whereas multidimensional frailty tools do not. It should be noted that these findings present limitations, given that studies exploring the prospective association of multidimensional frailty often lacked statistical power due to the smaller number of included studies in the meta-analyses.

Table 2 summarizes the meta-analyses of Vermeiren et al. [72], Yang et al. [73], and Cheng and Chan [74], separately presenting the prospective associations of physical and multidimensional frailty tools with various health-related adverse outcomes.

#### 3.5.2. One Construct to Predict Them All

Frailty has been widely conceptualized for its ability to predict a broad range of health-related adverse events, as it represents a state of vulnerability to such events. Based on this premise, frailty tools should be able to identify older adults at risk of negative outcomes, as discussed in the previous section. However, a closer look reveals some important challenges, particularly the assumption that frailty automatically implies a higher risk for any adverse health event.

In reality, the risk of specific adverse events depends on the particular characteristics present in each individual. An older adult with significant physical impairments, such as reduced balance, strength and coordination may be at higher risk of hip fracture, but not necessarily of cognitive decline or social isolation. Conversely, an older adult experiencing memory loss, disorientation, or attentional problems, may be more susceptible to conditions like Alzheimer’s disease but not at an increased risk of hip fracture; see Figure 9. Despite these distinct risk profiles, both individuals may be labeled as frail, which oversimplifies their unique vulnerabilities and reduces the accuracy of predictions.

Viewing frailty as a universal predictor of any adverse event is not only oversimplistic but also misleading. By gathering different risk profiles under a single construct, we lose accuracy for the possible prediction.

The previous section summarized the predictive validity of frailty instruments. However, there might be limitations in their results, mainly due to the potential bias in predicting events that are already being evaluated within the frailty scale. For example, there will appear an overestimation in the ability to predict an event, such as disability in BADL or IADL, if they are already being assessed within the frailty instrument.

Although several meta-analyses provide evidence for the prospective association of frailty with adverse events, the following points should be considered to improve accuracy and reduce bias in the prediction:**Plausibility of predictors:** Ensure that the factors identified as predictors are plausible triggers for the adverse event.**Specificity of predictors:** The predictors should be specific to the adverse event they aim to detect.**Validity and reliability of evaluations:** Ensure the assessment procedures, scoring categories, and weightings within the instrument are valid, and the instrument is reliable.**Exclude redundant items with the predictive event:** Remove items from the scale that are already part of the predictive event to avoid reiteration and overestimation bias in the prediction. For example, if the goal is to predict future limitations in IADL, the scale should avoid including items that assess limitations in IADL.

This situation highlights the importance of clinicians recognizing that scientific literature on the predictive validity of frailty may not always align with clinical practice. When evaluating a patient with a positive frailty score, clinicians should thoroughly assess the factors contributing to this score. Rather than simply suggesting a prognosis of multiple adverse health events, they should carefully consider which specific deficits in the patient are plausibly associated with the adverse events discussed in the scientific literature (Figure 9).

### 3.6. Not a Clinical Entity nor a Unique Etiology

A clinical entity is a concept developed by healthcare providers to identify and group individuals who share common characteristics. This concept arises from the clinician’s observations and reflections, enabling the clustering of patients with homogeneous features. As described by Guttentag [75], the clinical entity serves as a tool simplify the identification of patients, facilitating diagnosis, treatment, and comprehending the pathological mechanisms of the condition.

For example, a clinician may observe a group of patients presenting symptoms such as frequent urination, excessive thirst, unintentional weight loss, and elevated blood glucose levels. By identifying these consistent patterns and investigating the underlying pathophysiological mechanisms, the clinician can define diabetes mellitus as a clinical entity.

Similarly, in psychology, a practitioner may identify a clinical entity like depression by recognizing shared traits among individuals. These traits include episodes of sadness, loss of motivation, interest or pleasure in activities, difficulty concentrating, and changes in sleep or appetite. By recognizing these common characteristics, the clinician conceptualizes depression as a clinical entity.

Frailty has also been described as a clinical entity by several authors. Derived from the concept as a “clinical phenotype” by Fried et al. [1], several authors have stated: “*frailty is a clinical entity*” [76], “*a distinct clinical entity*” [77], “*the frailty syndrome is now a well-established clinical entity*” [78], or even considered by the WHO as a “*clinically recognizable state*”.

However, despite these statements, there are significant challenges that prevent frailty from being considered a true clinical entity. These challenges include:**Absence of a consistent pattern of common characteristics:** Unlike diabetes or depression, frailty does not emerge from the observation of a consistent and well-defined pattern of shared characteristics. Conditions like depression have been characterized through distinct and recurring symptoms, allowing for the development of validated assessment tools such as the Beck Depression Inventory, which gathers the traits identified in that clinical entity. In contrast, frailty characteristics are highly heterogeneous, with epidemiological studies highlighting these discrepancies.**General and abstract definitions:** In the case of diabetes, and depression, the observation of shared features forms the foundation for coining the name of those clinical entities. However, the definitions of frailty are often vague and lack specificity. This generalist approach fails to provide a clear and precise profile of identifiable patients.

In addition, literature has focused on exploring the etiology and pathophysiology of frailty [79]. Etiology and pathophysiology refer to the underlying factors and mechanisms that contribute to the onset of a condition, whether it is a clinical entity (including diseases or syndromes), or a specific event. However, as previously discussed, frailty does not fit in the category of clinical entity due to the diverse profiles identified as frail. Therefore, the etiology and pathophysiology of frailty vary depending on the specific characteristics that classify an individual as frailty, making it highly subject and profile-dependent.

Consequently, most studies examining the etiology of frailty have focused on specific profiles, such as the Fried phenotype or the 5-item FRAIL scale [80]. These studies have contributing factors which include sociodemographic variables (e.g., age, sex, and ethnicity), anthropometric and physical parameters, immune and endocrine biomarkers, lifestyle characteristics and psychological factors, among others. However, it is crucial to avoid generalizing these findings to the overall concept of frailty, given the heterogeneity of profiles classified under this term.

### 3.7. Frailty in the Context of Modern Medicine Paradigms

In the recent decades, following the emergence of evidence-based medicine, numerous paradigms have been developed in the field of medicine, focusing on the individualization, personalization, and precision of diagnostic methods, the establishment of the prognosis, and the design of appropriate treatments to address patients’ health-related issues. These paradigms, such as patient-centered care, person-focused care, precision medicine, and stratified medicine tailor medical decisions with the aim to be adapted to each individual.

While many of these paradigms have increasingly focused on molecular and genomic aspects, they remain fundamentally rooted in individual assessments based on patient’s clinical manifestations. From a clinical perspective, the medical interview, which encompasses the patient’s goals, needs, and capacities, is key in determining the appropriate type of evaluation to perform. Such evaluations must be both individualized and sufficiently precise to accurately identify the potentially influencing factors in the patient’s health status.

#### 3.7.1. Utility as a Screening Process Not for a Precise Evaluation

Frailty is a broadly defined construct which explores multiple health-related factors. This approach serves as a superficial evaluation, serving primarily as an initial framework to assess a patient’s condition. Therefore, in most cases, the final frailty score should not be overly emphasized. Instead, clinicians should focus more on the types of deficits leading to this punctuation. This initial screening can facilitate two essential objectives:Identify the need for more precise evaluations based on the deficits detected in frailty instruments.Guiding targeted interventions by addressing the specific deficits detected in the patient.

Frailty instruments are particularly useful in primary care settings, where clinicians can conduct a broad and rapid evaluation of potential factors affecting a patient’s health. However, the broad nature of frailty may present limitations of applicability in secondary care settings, where more specialized evaluations are required.

#### 3.7.2. Utility for Guiding Specific Interventions

Beyond their role in initial screening, frailty assessments can also help guide clinical decisions by identifying specific deficits, considering which additional evaluations to conduct, and stablish which specific interventions to conduct.

In the case of treatments, frailty instruments capture heterogeneous groups of subjects; therefore, not all individuals may benefit from the same intervention.

This issue is evidenced in certain trails. For example, exercise programs are widely recognized as a beneficial therapy for many older adults [81], yet it will not be universally effective for all frail individuals. A study by Pérez-Zepeda et al. [82] demonstrated that while an exercise significantly improved outcomes in frail subjects, not all participants benefited equally. This variability may be explained by the fact that frailty instruments assess multiple deficits across different health domains (as they employed a variant of the frailty index). As a result, some individuals may present characteristics that do not make them ideal candidates for a particular intervention.

This concept also applies to other interventions applied to frail subjects, such as nutritional [83], pharmacological management [84], or multimodal approaches [85]. While these interventions have demonstrated effectiveness, their success depends on appropriate patient selection, ensuring that treatments specifically target the deficits identified in each individual.

### 3.8. A Summary of Findings with the Proposal of a New Definition

After analyzing previous reviews of frailty definitions (poner refs), and expert panel consensus on the relationship between frailty and other constructs, we can identify several well-established points, as well as unresolved aspects of the construct.

This new definition of frailty begins with: “*Frailty is a state characterized by the accumulation of deficits*”. The use of the term “*state*” is intentional, as it avoids referring to frailty as a “*clinical entity*”, since it does not fit this classification. Likewise, we do not use the term “*clinical state*”, as this is also debatable due to the inclusion of socially related deficits.

Additionally, we do not specify which deficits are required for a patient to be considered frail, as observed across the different frailty instruments. This represents a weakness on the definition, as it does not clarify the type, number or severity of deficits necessary to classify an individual as frail.

Frailty is inherently based on the evaluation of specific deficits. While these deficits are not strictly defined, frailty can only arise on the presence of impairments in body functions or structures, diseases, social participation, education, rest and sleep, limitations in activities, health management, environmental and personal factors, among others, as discussed in “Section 3.3.1. Inconsistencies in Instruments’ Content”. Therefore, frailty relies on the assessment of multiple constructs. However, since these constructs are not explicitly defined, this represents a limitation in the definition of frailty. As discussed in “Section 3.4.4. Determining the Relationship Between Frailty and Other Constructs”, frailty cannot be based on the evaluation of a single construct, as this would make the concept redundant. Instead, frailty must assess multiple constructs to prove a distinctive assessment.

Another important concept discussed in the literature is the positioning of frailty within the health–death continuum. Several authors have proposed schematic models illustrating how frailty develops over time. These include *the cascade of functional decline* [19], or *the geriatric functional progression from independence to death* [86]. These schemes establish frailty as a prior state before the appearance of disability/dependency. This may be of use, as it identifies a situation of possible reversibility, as the capacity of reversibility may be limited when levels of dependency are severe. Although this approach is insightful, integrating frailty into this continuum is challenging due to several conceptual and operational issues:Frailty must be distinct from disability. Since disability is typically defined as limitations or dependency in activities, frailty should be identified in individuals before they reach that stage. To differentiate frailty from disability, two conditions should be met:
The individual should test positive on a frailty scale that does not include disability assessments (to provide a distinctiveness of the construct).The individual should test negative on a disability scale.Lack of a clear cut-off for defining disability. Disability is measured on a continuum, reflecting both the degree of dependency (e.g., independent, slightly dependent, and completely dependent), and the number of affected activities (e.g., toileting, bathing, and managing finances). This variability makes it difficult to establish a precise threshold that distinguishes frailty from disability.

Some individuals may meet frailty criteria while remaining fully independent, while others may experience mild to severe limitations in activities, therefore presenting both characteristics of frailty and disability. The lack of a standardized cut-off point makes this classification complex. Therefore, we argue that frailty should not be defined strictly within the health–death continuum, as this approach creates operational conflicts with the presence of disability.

One final issue to address, is the “vulnerability” or “risk” for “future health-related adverse events” aimed to be assessed in frailty. Many definitions propose that frailty is a state of vulnerability or risk for future health-related adverse events. As discussed in the previous section, establishing the risk is a complex task, and even more, when the variability in the assessments in frailty scales is high, and so is the plausibility of the future adverse event detected.

Therefore, due to plausibility issues, we think that the definition should avoid establishing a future risk event, as a frailty patient cannot present multiple and undefined adverse risks events. One patient based on its deficits will present certain risks, while another patient may present another type of risk. Therefore, we do not count with

One last issue to address is how “*vulnerability*” or “*risk*” *for future health-related adverse events* should be applied in the definition of frailty. Many definitions describe frailty as a state of vulnerability to future adverse events. However, as discussed in the previous section, establishing this risk is highly complex due to the wide variability in frailty assessment tools and the lack of specificity in defining which adverse events a frail patient may be at risk for.

Given these plausibility issues, we argue that the definition of frailty should avoid assuming a generalized future risk for multiple undefined adverse events. A frail patient does not face all possible health risks equally; rather, the risks they face depend on the specific deficits they present. For example, a patient with significant physical impairments may be at higher risk for falls and fractures, whereas a patient with cognitive decline may be more vulnerable to dementia-related complications.

Therefore, the definition of frailty should move away from a broad, nonspecific risk categorization and instead focus on identifying plausible and specific risks based on the patient’s individual profile. This approach enhances both clinical relevance and predictive accuracy, preventing frailty from being used as an indiscriminate predictor of all possible health-related adverse events.

Based on the information discussed we propose the following definition of frailty:

“*Frailty is a state characterized by the accumulation of deficits across various health domains. It is not a clinical entity and does not present a clear etiology. Frailty should not be considered as a precursor to disability, as it is a distinct construct rather than a pre-disability stage. Additionally, disability exists on continuum without clear thresholds, and both conditions may coexist. Its identification should be based on the evaluation of multiple rather than a single health domain to provide a distinctive evaluation. While frailty may be related to future adverse health events, any potential risk should be assessed based on the individual’s specific deficits, rather than considering frailty as a multiple risk indicator.*”

## 4. Discussion

### 4.1. Conceptual and Operational Heterogeneity

This review highlights the significant variability in the conceptualization and operationalization of frailty, which impedes a unified understanding. Historically, the term “frailty” was introduced through the different perspectives of Fried et al. [1] and Mitnitski et al. [6], and currently lacks a unified understanding with a well-established theoretical framework. Future research must establish a proper theoretical framework to ensure the construct’s utility. A unified theoretical framework for frailty is essential to (1) providing a consistent foundation for research and clinical practice, reducing the confusion and variability in the understanding of the construct; (2) properly establishing its boundaries with other constructs, justifying its utility; (3) enhancing the coherence across the development of frailty instruments; and (4) targeting and designing interventions to reduce its impact.

### 4.2. Content Validity of Frailty

The review also reveals significant inconsistencies in the content, assessment procedures, scoring systems, and weightings of current frailty instruments, affecting the validity and reliability of the assessments. Moreover, many instruments do not undergo a content validity process, further questioning their validity. A standardized content validity process should be mandated to improve the quality of frailty instruments. This involves: (1) establishing items in line with the developed conceptual framework; (2) ensuring all items are evaluated by an expert panel for their comprehension, coherence (with the construct), and relevance; (3) conducting a deep examination of the validity of the assessment procedures, response and scoring systems, and its weightings within the instrument, and (4) including the participation of clinicians and older adults for further perspectives on the content of the instrument.

### 4.3. Distinctiveness and Predictive Validity of Frailty

As discussed, the concept of frailty often overlaps with disability and morbidity, questioning its distinctiveness and predictive utility. The lack of specificity in frailty definitions could lead to its redundancy with other constructs, diluting its clinical utility. Additionally, the predictive validity of frailty for adverse health outcomes varies, potentially due to inherent biases in the instruments, such as: (1) the plausibility, and specificity of predictor factors—for example, an increase in fall risk in the presence of sarcopenia could easily be expected [87]; meanwhile, it could be more difficult to detect a relationship between cognitive impairment and fracture risk, although they are related [88]; (2) the validity and reliability of evaluations; and (3) avoiding predicting events or constructs already assessed within the instrument (e.g., avoid including evaluations of limitations in ADL when the event aimed to predict is limitations in ADL).

By addressing these issues, the frailty construct can be more accurately and consistently assessed, improving its utility in predicting health outcomes and guiding interventions for older adults. These recommendations should foster a more coherent understanding and application of frailty in geriatric care.

### 4.4. Integration of Multidimensional and Contextual Factors in Frailty Assessment

Although frailty is typically viewed through a physical or clinical lens, emerging evidence underscores the importance of considering social, environmental, and psychological factors in the development and progression of patient’s vulnerability [89,90]. These multidimensional aspects, such as social support, living conditions, and emotional well-being, can significantly influence the severity and outcomes of frailty, yet are often overlooked in current assessment tools. For instance, socioeconomic status, access to healthcare, and social networks have been shown to affect frailty levels and the ability to recover from adverse health outcomes [91].

Incorporating these factors into frailty assessments would allow for a more comprehensive evaluation of older adults’ vulnerabilities, moving beyond physical deficits to include biopsychosocial dimensions. Doing so would not only improve the accuracy of frailty predictions but also enable more personalized interventions that account for the broader social determinants of health [19]. For example, individuals with strong social support systems may be better equipped to manage frailty, even with similar physical impairments, compared with those who are socially isolated [5].

This multidimensional approach would also address health disparities across diverse populations, highlighting how frailty manifests differently depending on environmental and societal contexts [90]. Future research should prioritize the development of comprehensive frailty models that integrate these contextual factors to enhance the predictive validity and clinical utility of frailty assessments [19].

## 5. Conclusions

Frailty remains a complex and debated construct in geriatrics, marked by diverse definitions and a lack of consensus. The historical divergence in its conceptualization has contributed to ongoing confusion. Despite numerous efforts, a unified theoretical framework for frailty has not been established, impeding consistent research, assessment, and intervention strategies. This review highlights significant inconsistencies in frailty assessment tools, emphasizing the need for a standardized understanding of frailty, along with proper content validity processes.

Frailty overlaps not only with disability or morbidity, but also with other constructs, hindering its distinctiveness. Frailty is a construct which can only exist under the presence of other underlying constructs. The variability in scales content, scoring, and weighting systems results in the detection of heterogeneous samples, all labeled under the “frailty” tag. This also leads to difficulties in the interpretation of the responsiveness of these scales. This approach conflicts with modern medicine paradigms based on individualization, and precision. While frailty scales may be useful in primary care, they present challenges for providing thorough assessments in secondary care. Current research suggests that frailty is not a clinical entity.

Moreover, the predictive utility of frailty remains unclear due to inconsistencies within and between instruments. Establishing a unified theoretical framework along with proper development of instruments can enhance the predictive validity and clinical utility of frailty, ultimately improving care for older adults.

## Figures and Tables

**Figure 1 jcm-14-01808-f001:**
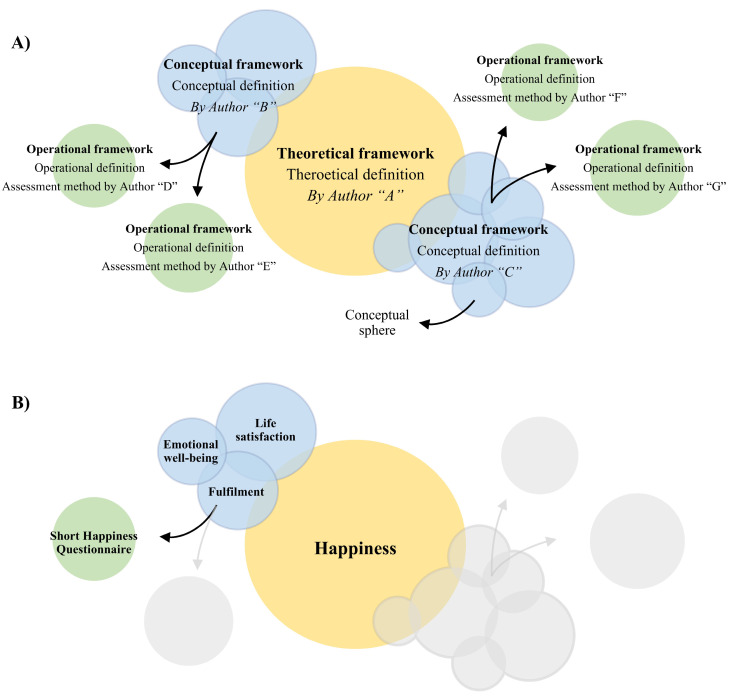
(**A**) The process of defining a construct within a theoretical framework, conceptualizing the key dimensions (conceptual spheres) that compose the construct, and finally operationalizing it by developing an instrument to measure each dimension. (**B**) An example using the construct of “happiness”, which is first defined theoretically, then broken down into conceptual spheres, and ultimately measured through the development of an instrument.

**Figure 2 jcm-14-01808-f002:**
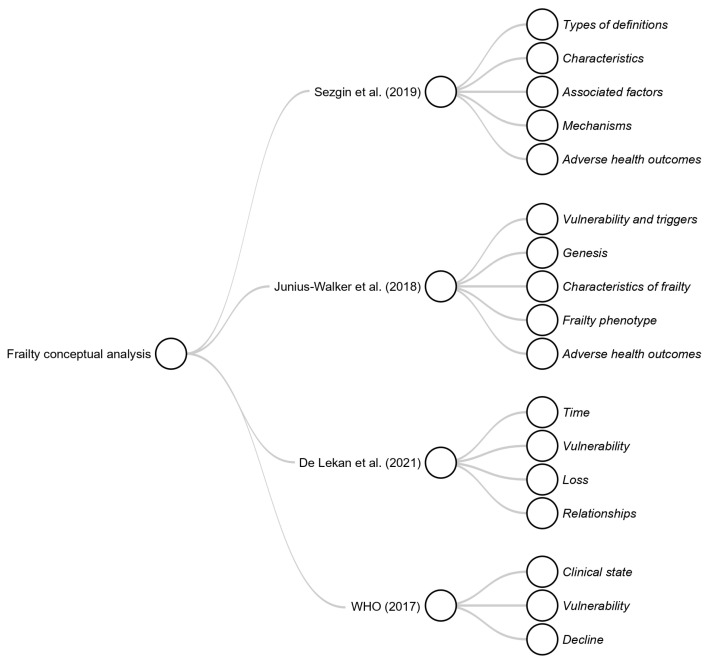
An overview of the conceptual domains of frailty identified in the literature. It integrates key themes from systematic reviews which analyze frailty definitions, highlighting the different perspectives used to define and understand frailty. Additionally, the figure incorporates the World Health Organization’s [15] conceptual definition of frailty.

**Figure 3 jcm-14-01808-f003:**
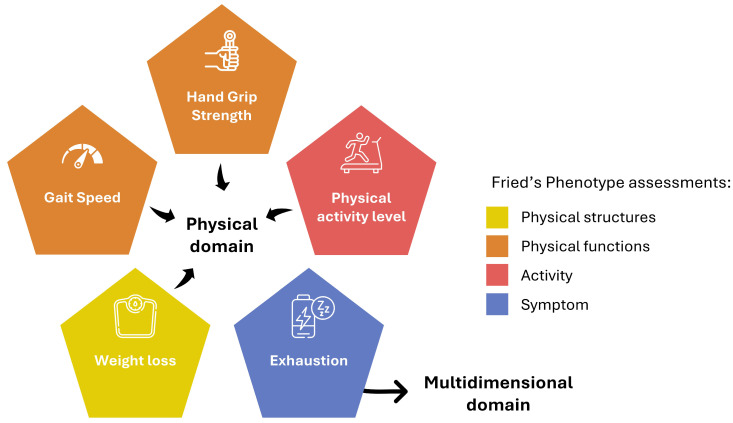
Our classification of the assessment items from the frailty phenotype model [1], highlighting that despite being labeled as a measure of “physical frailty”, it does not assess exclusively physical aspects. Its assessments include physical structures (weight loss), physical functions (hand grip strength, gait speed), activities (physical activity level), and symptoms (exhaustion/fatigue), the latter being a multidimensional construct. This classification reflects how the term “physical frailty” has been somewhat misinterpreted in the literature when applied to certain frailty assessment tools.

**Figure 4 jcm-14-01808-f004:**
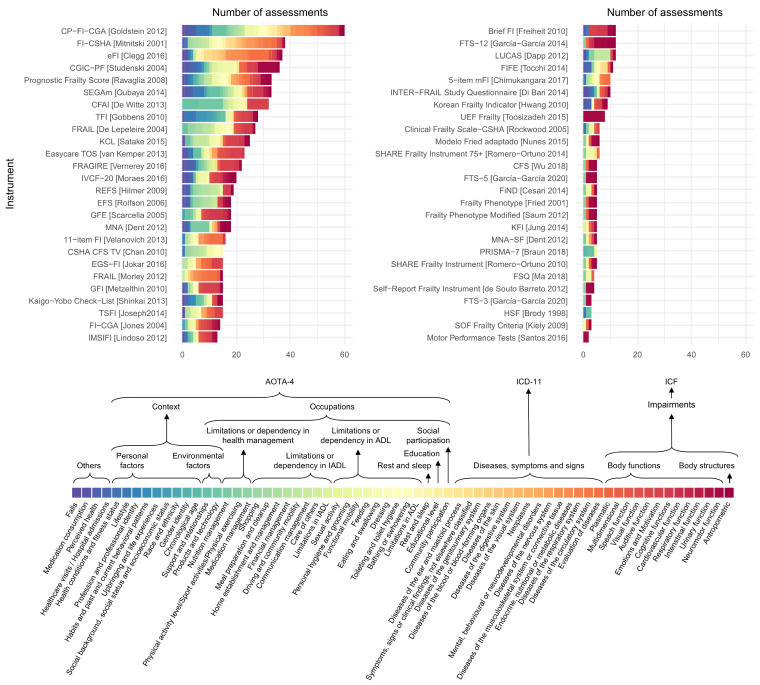
A detailed content analysis of frailty instruments’ assessments based on specific domains of evaluation proposed by the International Classification of Functioning, Disability and Health (ICF), the International Classification of Diseases 11th Edition (ICD-11), and the Occupational Therapy Practice Framework: Domain and Process 4th Edition by the American Occupational Therapy Association (AOTA-4).

**Figure 5 jcm-14-01808-f005:**
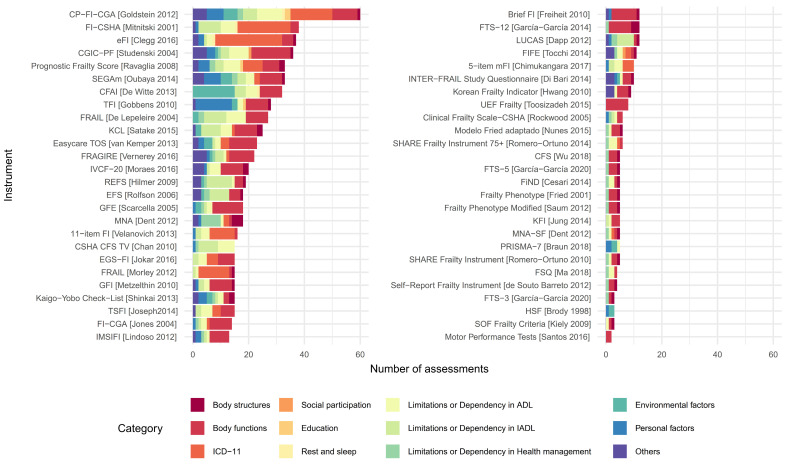
A broader and general content analysis of frailty instruments’ assessments based on broad domains of evaluation following the ICF, ICD-11 and AOTA-4.

**Figure 6 jcm-14-01808-f006:**
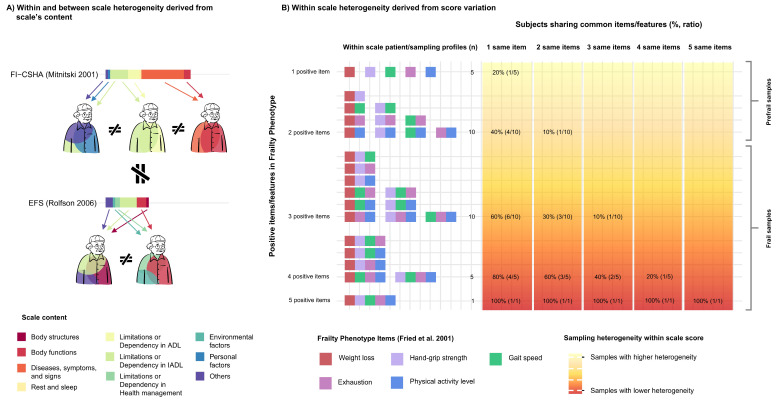
(**A**) How diverse content of an instrument can lead to the identification of heterogeneous patient profiles. (**B**) As frailty scores decrease, the identified profiles become more diverse, as individuals share fewer common traits. Conversely, as frailty scores increase, the identified profiles become more similar, as individuals share more common characteristics.

**Figure 7 jcm-14-01808-f007:**
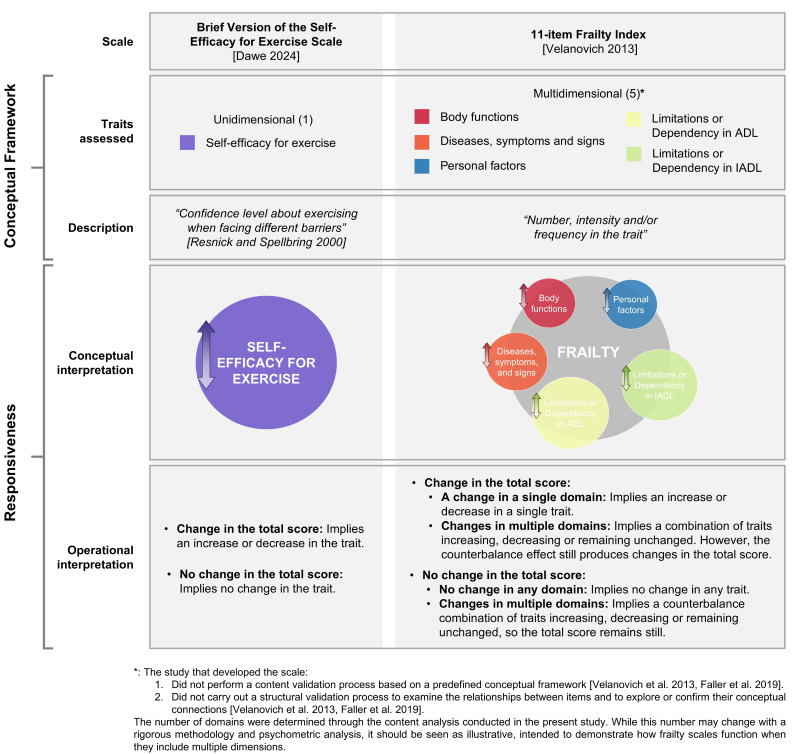
The complexity of interpreting frailty scales’ responsiveness. Unlike unidimensional assessment tools where a change in total score directly reflects a change in the trait being measured, frailty scales allow for different scenarios. A shift in total score may result from changes in a single or a combination of domains, while the total score can also remain unchanged despite underlying trait fluctuations due to compensatory effects across domains.

**Figure 8 jcm-14-01808-f008:**
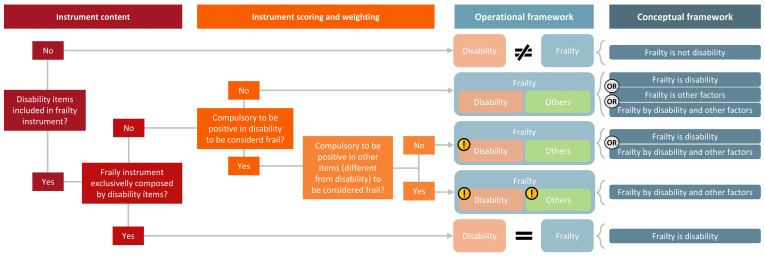
The process of how two constructs can be conceptually related based on the operational framework of the instrument. The content of the instrument, along with its weighting and scoring system, determines the potential relationship between constructs.

**Figure 9 jcm-14-01808-f009:**
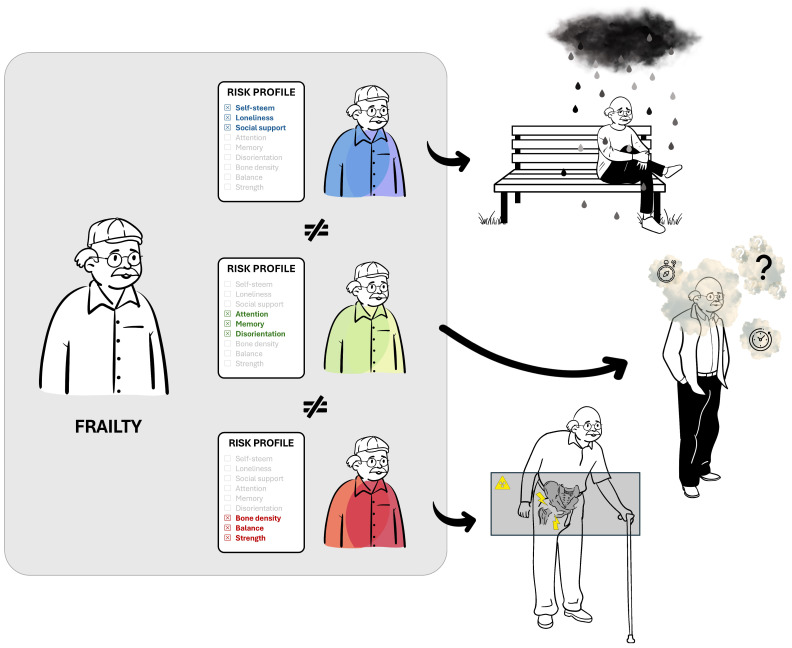
How an older adult can be classified as frail due to different underlying factors, such as physical, cognitive or psychosocial impairments. Each profile represents a distinct pathway to frailty, meaning that a patient identified as frail in one domain may not necessarily be at risk for adverse events linked to another domain (e.g., cognitive decline or hip fracture) due to the lack of a plausible connection. This challenges many frailty definitions that state that a frail individual is inherently at risk for multiple health-related adverse events.

**Table 1 jcm-14-01808-t001:** Prospective association among frailty assessment tools (physical and multidimensional tools grouped).

Population	Study	Groups	Result	Frailty Tools	Follow-Up (OR)	Studies *(comp.)* (OR)	OR (95%CI)	Follow-Up (HR/RR)	Studies *(comp.)* (HR/RR)	HR/RR (95%CI)
**Mortality**										
Community-dwelling older adults ≥65 years	Vermeiren et al., 2016 [72]	F and PF vs. R	** Risk **	All	1–7 years	8 *(24)*	**2.34 (1.77, 3.09) ***	0.83–10 years	15 *(37)*	**1.83 (1.68, 1.98) ***
		F vs. R	** Risk **	All	1–7 years	8 *(18)*	**2.55 (1.76, 3.70) ***	0.83–10 years	15 *(25)*	**2.01 (1.82, 2.22) ***
		PF vs. R	** Risk **	All	2–7 years	3 *(6)*	**1.76 (1.36, 2.28) ***	0.83–10 years	9 *(12)*	**1.47 (1.32, 1.64) ***
**Hospitalization**										
Community-dwelling older adults ≥65 years	Vermeiren et al., 2016 [72]	F and PF vs. R	** Risk **	All	0.83–5.75 years	8 *(19)*	**1.82 (1.53, 2.15) ***	1–7 years	3 *(13)*	**1.18 (1.10, 1.28) ***
		F vs. R	** Risk **	All	0.83–5.75 years	8 *(14)*	**1.97 (1.58, 2.46) ***	1–7 years	3 *(7)*	**1.23 (1.07, 1.40) ***
		PF vs. R	** Risk **	All	0.83–5.75 years	4 *(5)*	**1.53 (1.19, 1.96) ***	1–7 years	2 *(6)*	**1.15 (1.06, 1.24) ***
**Institutionalization**										
Community-dwelling older adults ≥65 years	Vermeiren et al., 2016 [72]	F and PF vs. R	** Risk **	All	-	-	-	1–8 years	3 *(12)*	**1.65 (1.48, 1.84) ***
		F vs. R	** Risk **	All	1–4 years	2 *(4)*	**1.69 (1.02, 2.81) ***	1–8 years	3 *(8)*	**1.67 (1.47. 1.89) ***
		PF vs. R	** Risk **	All	-	-	-	1 year	1 *(4)*	**1.55 (1.26, 1.91) ***
**ED visits**										
Community-dwelling older adults ≥65 years	Vermeiren et al., 2016 [72]	F and PF vs. R	NA	All	-	-	-	-	-	-
		F vs. R	NA	All	-	-	-	-	-	-
		PF vs. R	NA	All	-	-	-	-	-	-
**Disability in BADL**										
Community-dwelling older adults ≥65 years	Vermeiren et al., 2016 [72]	F and PF vs. R	** Risk **	All	1–5.75 years	8 *(18)*	**2.05 (1.73, 2.44) ***			
		F vs. R	** Risk **	All	1–5.75 years	8 *(13)*	**2.13 (1.76, 2.59) ***	-	-	-
		PF vs. R	NA	All	-	-	-	-	-	-
**Disability in IADL**										
Community-dwelling older adults ≥65 years	Vermeiren et al., 2016 [72]	F and PF vs. R	** Risk **	All	0.83–7 years	7 *(19)*	**2.73 (2.19, 3.42) ***	-	-	-
		F vs. R	** Risk **	All	0.83–7 years	7 *(12)*	**3.06 (2.13, 4.39) ***	-	-	-
		PF vs. R	NA	All	-	-	-	-	-	-
**Physical limitation**										
Community-dwelling older adults ≥65 years	Vermeiren et al., 2016 [72]	F and PF vs. R	** Risk **	All	4–5 years	2 *(9)*	**2.58 (1.85, 3.62) ***	-	-	-
		F vs. R	** Risk **	All	4–5 years	2 *(5)*	**3.63 (2.14, 6.16) ***	-	-	-
		PF vs. R	** Risk **	All	4–5 years	2 *(4)*	**1.81 (1.41, 2.33) ***	-	-	-
**Dependency**										
Community-dwelling older adults ≥65 years	Vermeiren et al., 2016 [72]	F and PF vs. R	NA	All	-	-	-	-	-	-
		F vs. R	** Risk **	All	-	-	-	3 years	1 *(2)*	**1.32 (1.19, 1.47) ***
		PF vs. R	NA	All	-	-	-	-	-	-
**Falls**										
Older adults ≥65 years	Yang (2023)	F vs. R	** Risk **	All	-	-	-	0.33–11 years	29 *(29)*	**1.48 (1.27, 1.73) ***
Community-dwelling older adults ≥65 years	Vermeiren et al., 2016 [72]	F and PF vs. R	** Risk **	All	0.83–9 years	3 *(8)*	**1.70 (1.18, 2.44) ***	0.83–8 years	3 *(9)*	**1.24 (1.12, 1.37) ***
		F vs. R	** Risk **	All	0.83–9 years	3 *(5)*	**2.06 (1.28, 3.34) ***	0.83–7 years	3 *(5)*	**1.34 (1.14, 1.58) ***
		PF vs. R	NA	All	-	-	-	-	-	-
**Fractures**										
Community-dwelling older adults ≥65 years	Vermeiren et al., 2016 [72]	F and PF vs. R	NA	All	-	-	-	-	-	-
		F vs. R	NA	All	-	-	-	-	-	-
		PF vs. R	NA	All	-	-	-	-	-	-
**Cognitive decline**										
Community-dwelling older adults ≥65 years	Vermeiren et al., 2016 [72]	F and PF vs. R	NA	All	-	-	-	-	-	-
		F vs. R	NA	All	-	-	-	-	-	-
		PF vs. R	NA	All	-	-	-	-	-	-
**Body composition**										
Community-dwelling older adults ≥65 years	Vermeiren et al., 2016 [72]	F and PF vs. R	NA	All	-	-	-	-	-	-
		F vs. R	NA	All	-	-	-	-	-	-
		PF vs. R	NA	All	-	-	-	-	-	-
**Life satisfaction**										
Community-dwelling older adults ≥65 years	Vermeiren et al., 2016 [72]	F and PF vs. R	NA	All	-	-	-	-	-	-
		F vs. R	NA	All	-	-	-	-	-	-
		PF vs. R	NA	All	-	-	-	-	-	-

***** and Bold values represent statistically significant risk. Comp., number of pairwise comparisons; F, frail; PF, prefrail; R, robust.

**Table 2 jcm-14-01808-t002:** Prospective association between physical and multidimensional frailty assessment tools.

Population	Groups	Risk Between Tool Types	Tool Type	Specific Tool	Study	Risk	Follow-Up (OR)	Studies *(comp.)* (OR)	OR (95%CI)	Follow-Up (HR/RR)	Studies *(comp.)* (HR/RR)	HR/RR (95%CI)
**Mortality**												
Community-dwelling older adults ≥65 years	F and PF vs. R	** Similar ᶲ **	Physical	Several	Vermeiren et al., 2016 [72]	** Risk **	2–7 years	4 *(12)*	**2.58 (1.83, 3.64) ***	0.83–10 years	12 *(25)*	**1.70 (1.49, 1.95) ***
			Mult-CD	FI	Vermeiren et al., 2016 [72]	** Risk **	-	-	-	4–5.17 years	4 *(7)*	**3.64 (1.72, 7.72) ***
			Mult-Non-CD	Several	Vermeiren et al., 2016 [72]	** Risk **	1–7 years	5 *(11)*	**2.13 (1.38, 3.29) ***	-	-	-
	F vs. R	NA	Physical			NA						
			Mult-CD	FI	Vermeiren et al., 2016 [72]	** Risk **	2 years	1 *(1)*	**1.85 (1.30, 2.63) ***	-	-	-
			Mult-Non-CD	Various	Vermeiren et al., 2016 [72]	** Risk **	-	-	-	3–8 years	4 *(5)*	**1.32 (1.22, 1.43) ***
**Hospitalization**												
Community-dwelling older adults ≥65 years	F and PF vs. R	** Similar ᶲ **	Physical	Various	Vermeiren et al., 2016 [72]	** Risk **	0.83–5.75 years	5 *(11)*	**1.83 (1.47, 2.28) ***	1–7 years	3 *(7)*	**1.16 (1.06, 1.27) ***
			Mult-CD	Various	Vermeiren et al., 2016 [72]	** Risk **	-	-	-	1 year	1 *(4)*	**1.20 (1.02, 1.41) ***
			Mult-Non-CD	CHESS	Vermeiren et al., 2016 [72]	** Risk **	-	-	-	1 year	1 *(2)*	**1.26 (1.12, 1.42) ***
	F vs. R	NA	Physical			NA						
			Mult-CD			NA						
			Mult-Non-CD	Various	Vermeiren et al., 2016 [72]	** Risk **	1–4 years	4 *(8)*	**1.84 (1.35, 2.51) ***	-	-	-
**Institutionalization**												
Community-dwelling older adults ≥65 years	F and PF vs. R	** Similar ᶲ **	Physical	Frailty Phen.	Vermeiren et al., 2016 [72]	** Risk **	-	-	-	1 year	1 *(2)*	**1.82 (1.26, 2.63) ***
			Mult-CD	Various	Vermeiren et al., 2016 [72]	** Risk **	-	-	-	1–5 years	2 *(5)*	**2.30 (1.54, 3.43) ***
			Mult-Non-CD	Various	Vermeiren et al., 2016 [72]	** Risk **	-	-	-	1–8 years	3 *(5)*	**1.44 (1.28, 1.62) ***
	F vs. R	** Similar ᶲ **	Physical	mSOF	Vermeiren et al., 2016 [72]	** No **	4 years	1 *(1)*	2.53 (0.71, 9.02)	-	-	-
			Mult-CD			NA						
			Mult-Non-CD	Various	Vermeiren et al., 2016 [72]	** No **	1–4 years	2 *(3)*	1.62 (0.93, 2.81)	-	-	-
**ED visits**												
Community-dwelling older adults ≥65 years	F and PF vs. R	** Different ^‡^ **	Physical	Various	Vermeiren et al., 2016 [72]	** Risk **	0.83 years	1 *(4)*	**2.16 (1.39, 3.37) ***	-	-	**-**
			Mult-CD	DAI	Vermeiren et al., 2016 [72]	** No **	-	-	-	0.08 years	1 *(3)*	1.03 (0.82, 1.29)
			Mult-Non-CD			NA						
	F vs. R	NA	Physical	Various	Vermeiren et al., 2016 [72]	** Risk **	0.83 years	1 *(2)*	**3.24 (1.92, 5.45) ***	-	-	-
			Mult-CD			NA						
			Mult-Non-CD			NA						
	PF vs. R	NA	Physical	Various	Vermeiren et al., 2016 [72]	** Risk **	0.83 years	1 *(2)*	**1.69 (1.39, 2.72) ***	-	-	-
**Disability in BADL**												
Community-dwelling older adults ≥65 years	F and PF vs. R	NA	Physical	Various	Vermeiren et al., 2016 [72]	** Risk **	1–5 years	6 *(12)*	**2.11 (1.61, 2.76) ***	3–8 years	2 *(6)*	**1.62 (1.50, 1.76) ***
			Mult-CD			NA						
			Mult-Non-CD			NA						
	F vs. R	** Similar ᶲ **	Physical	Various	Vermeiren et al., 2016 [72]	** Risk **	-	-	-	3–8 years	3 *(4)*	**1.67 (1.45, 1.92) ***
			Mult-CD			NA						
			Mult-Non-CD	Various (OR) Physical Scale overall (HR/RR)	Vermeiren et al., 2016 [72]	** Risk **	1–4 years	3 *(6)*	**1.80 (1.45, 2.22) ***	8 years	1 *(1)*	**1.59 (1.21, 2.09) ***
	PF vs. R	NA	Physical	Various (OR) Frailty Phen. (HR/RR)	Vermeiren et al., 2016 [72]	** Risk **	3–4 years	4 *(5)*	**1.86 (1.35, 2.56) ***	3–7 years	1 *(2)*	**1.59 (1.44, 1.75) ***
**Disability in IADL**												
Community-dwelling older adults ≥65 years	F and PF vs. R	NA	Physical	Various	Vermeiren et al., 2016 [72]	** Risk **	0.83–7 years	6 *(16)*	**2.81 (2.20, 3.58) ***	-	-	-
			Mult-CD			NA						
			Mult-Non-CD			NA						
	F vs. R	NA	Physical			NA						
			Mult-CD			NA						
			Mult-Non-CD	Various	Vermeiren et al., 2016 [72]	** Risk **	1 year	1 *(3)*	**2.33 (1.68, 3.23) ***	-	-	-
	PF vs. R	NA	Physical	Variouos	Vermeiren et al., 2016 [72]	** Risk **	0.83–4 years	4 *(7)*	**2.30 (1.95, 2.72) ***	-	-	-
**Physical limitation**												
Community-dwelling older adults ≥65 years	F and PF vs. R	NA	Physical	Frailty Phen.	Vermeiren et al., 2016 [72]	** Risk **	-	-	-	3–7 years	1 *(4)*	**1.46 (1.37, 1.56) ***
			Mult-CD			NA						
			Mult-Non-CD			NA						
	F vs. R	NA	Physical	Frailty Phen.	Vermeiren et al., 2016 [72]	** Risk **	-	-	-	3–7 years	1 *(2)*	**1.42 (1.25, 1.61) ***
			Mult-CD			NA						
			Mult-Non-CD			NA						
	PF vs. R	NA	Physical	Frailty Phen.	Vermeiren et al., 2016 [72]	** Risk **	-	-	-	3–7 years	1 *(2)*	**1.48 (1.33, 1.66) ***
**Dependency**												
Community-dwelling older adults ≥65 years	F and PF vs. R	NA	Physical			NA						
			Mult-CD			NA						
			Mult-Non-CD			NA						
	F vs. R	NA	Physical			NA						
			Mult-CD			NA						
			Mult-Non-CD			NA						
	PF vs. R	NA	Physical			NA						
**Falls**												
Community-dwelling older adults ≥65 years	F and PF vs. R	NA	Physical	Various	Vermeiren et al., 2016 [72]	** Risk **	0.83–9 years	3 *(7)*	**1.72 (1.16, 2.54) ***	0.83–7 years	2 *(8)*	**1.26 (1.12, 1.41) ***
			Mult-CD			NA						
			Mult-Non-CD			NA						
	F vs. R	** Different ^‡^ **	Physical	Various	Cheng et al., 2017 [74]	** Risk **	1–3 years	7 *(7)*	**2.50 (1.58–3.96) ***	1.5–3 years	3 *(3)*	**1.48 (1.07, 2.04) ***
				Frailty Phen.	Cheng et al., 2017 [74]	** Risk **	1–3 years	6 *(6)*	**2.37 (1.43, 3.94) ***	1.5–3 years	3 *(3)*	**1.38 (1.10, 1.75) ***
				SOF index	Cheng et al., 2017 [74]	** Risk **	1 year	3 *(3)*	**2.73 (2.11, 3.53) ***	1.5 years	1 *(1)*	**2.19 (1.19, 4.03) ***
			Mult-CD			NA						
			Mult-Non-CD	CSBA	Vermeiren et al., 2016 [72]	** No **	4 years	1 *(1)*	1.49 (0.69, 3.22)	8 years	1 *(1)*	1.21 (0.95, 1.53)
	PF vs. R	NA	Physical	Various	Vermeiren et al., 2016 [72]	** Controversy **	0.83–9 years	2 *(3)*	1.31 (0.89, 1.93)	0.83–7 years	2 *(4)*	**1.17 (1.05, 1.30) ***
					Cheng et al., 2017 [74]	** Risk **	1–3 years	7 *(7)*	**1.47 (1.22, 1.79) ***	1.5–3 years	3 *(3)*	**1.17 (1.02, 1.34) ***
				Frailty Phen.	Cheng et al., 2017 [74]	** Risk **	1–3 years	6 *(6)*	**1.46 (1.18, 1.82) ***	1–3 years	3 *(3)*	1.12 (0.99, 1.27)
				SOF index	Cheng et al., 2017 [74]	** Risk **	1 year	3 *(3)*	**1.43 (1.24, 1.65) ***	1.5 years	1 *(1)*	**1.62 (1.14, 2.31) ***
Older adults ≥65 years	F vs. R	** Different ^‡^ **	Physical	Frailty Phen.	Yang (2023) [73]	** Risk **	-	-	-	1–7 years	9 *(9)*	**1.32 (1.17, 1.48) ***
				FRAIL scale	Yang (2023) [73]	** Risk **	-	-	-	0.5–3 years	6 *(6)*	**1.82 (1.36, 2.43) ***
				SOF index	Yang (2023) [73]	** Risk **	-	-	-	1–10 years	3 *(3)*	**1.54 (1.10, 2.16) ***
			Mult-CD	FI	Yang (2023) [73]	** No **	-	-	-	1–6 years	2 *(2)*	0.91 (0.52, 1.57)
				e-FI	Yang (2023) [73]	** No **	-	-	-	9–11 years	2 *(2)*	1.52 (0.65, 3.56)
			Mult-Non-CD	CFS	Yang (2023) [73]	** No **	-	-	-	0.33–1 year	3 *(3)*	1.55 (0.76, 3.16)
**Fractures**												
Community-dwelling older adults ≥65 years	F and PF vs. R	NA	Physical	Various	Vermeiren et al., 2016 [72]	** Risk **	3–4 years	2 *(3)*	**3.35 (1.18, 9.55) ***	0.83–9 years	3 *(10)*	**1.37 (1.21, 1.54) ***
			Mult-CD			NA						
			Mult-Non-CD			NA						
	F vs. R	** Different ^‡^ **	Physical	Various	Vermeiren et al., 2016 [72]	** Risk **	-	-	-	0.83–9 years	3 *(5)*	**1.59 (1.27, 2.00) ***
			Mult-CD			NA						
			Mult-Non-CD	CSBA	Vermeiren et al., 2016 [72]	** No **	3 years	1 *(1)*	1.76 (0.99, 3.13)	-	-	-
	PF vs. R	NA	Physical	Various	Vermeiren et al., 2016 [72]	** Risk **	-	-	-	0.83–9 years	3 *(5)*	**1.18 (1.21, 1.29) ***
**Cognitive decline**												
Community-dwelling older adults ≥65 years	F and PF vs. R	NA	Physical	Frailty Phen.	Vermeiren et al., 2016 [72]	** Risk **	-	-	-	4–5 years	1 *(3)*	**1.47 (1.23, 1.76) ***
			Mult-CD			NA						
			Mult-Non-CD			NA						
	F vs. R	NA	Physical			NA						
			Mult-CD			NA						
			Mult-Non-CD			NA						
	PF vs. R	NA	Physical			NA						
**Body composition**												
Community-dwelling older adults ≥65 years	F and PF vs. R	NA	Physical	Frailty Phen.	Vermeiren et al., 2016 [72]	** No **	5.17 years	1 *(2)*	1.95 (0.73, 5.19)	-	-	-
			Mult-CD			NA						
			Mult-Non-CD			NA						
	F vs. R	NA	Physical	Frailty Phen.	Vermeiren et al., 2016 [72]	** Risk **	5.17 years	1 *(1)*	**3.41 (1.57, 7.41) ***	-	-	-
			Mult-CD			NA						
			Mult-Non-CD			NA						
	PF vs. R	NA	Physical	Frailty Phen.	Vermeiren et al., 2016 [72]	** No **	5.17 years	1 *(1)*	1.25 (0.83, 1.88)	-	-	-
**Life satisfaction**												
Community-dwelling older adults ≥65 years	F and PF vs. R	NA	Physical			NA						
			Mult-CD			NA						
			Mult-Non-CD	Brief Frailty Instrument	Vermeiren et al., 2016 [72]	** Risk **	1 year	1 *(2)*	**2.62 (1.64, 5.13) ***	-	-	-
	F vs. R	NA	Physical			NA						
			Mult-CD			NA						
			Mult-Non-CD	Brief Frailty Instrument	Vermeiren et al., 2016 [72]	** Risk **	1 year	1 *(1)*	**3.88 (1.61, 9.35) ***	-	-	-
	PF vs. R	NA	Mult-Non-CD	Brief Frailty Instrument	Vermeiren et al., 2016 [72]	** Risk **	1 year	1 *(1)*	1.94 (0.94, 4.00)	-	-	-

ᶲ Physical and multidimensional frailty tools present both a positive risk for detecting the health-related adverse outcome prospectively. ^‡^ Physical frailty tools present a positive risk for detecting the health-related adverse outcome, while multidimensional frailty tools do not present an association with the outcome. ***** and Bold values represent statistically significant risk. Brief Frailty Instrument, the tool proposed by Rockwood et al., 1999; CFS, The 9-point Clinical Frailty Scale proposed by Rockwood et al., 2005; CSBA, The 9-item index proposed by Ravaglia et al., 2008 based on the Conselice Study of Brain Ageing population-based study; Comp., number of pairwise comparisons; e-FI, the 36-item electronic frailty index proposed by Clegg et al., 2017; F, frail; FI, the frailty index proposed by Mitnitski et al., 2001; FRAIL scale; the 5-item tool proposed by Morley et al., 2012; Frailty Phen., the original Fried frailty phenotype or modified assessments based on the original scale; Mult-CD, multidimensional frailty tools based on Cumulative-Deficit models; Mult-Non-CD, multidimensional frailty tools not based on Cumulative-Deficit models; PF, prefrail; R, robust; SOF index, the Study of Osteoporotic Fractures Index of 3-items proposed by Ensrud et al., 2008.

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
