# Peer review of "Frailty in Geriatrics: A Critical Review with Content Analysis of Instruments, Overlapping Constructs, and Challenges in Diagnosis and Prognostic Precision"

_jcm, 2025, doi:10.3390/jcm14061808_

Round 1

Reviewer 1 Report

Comments and Suggestions for Authors

Great paper! It provides a good overview and multiple perspectives on frailty syndrome, which remains a vague, poorly understood, and not yet well-established concept. However, there are some remarks that the authors should address:

  1. The authors should discuss the definition of frailty as an accumulation of deficits and provide insights into tools like the Rockwood Frailty Index and similar indices.
  2. They should elaborate on the associations of frailty with specific diseases. For example, frailty is observed in up to 50% or more of patients with heart failure (HF) but is less prevalent in other chronic diseases. This could offer valuable context.
  3. The manuscript should delve deeper into the relationship between frailty and sarcopenia, including underlying mechanisms. Why is frailty distinct from sarcopenia? Can a patient be frail but not sarcopenic? Conversely, can a robust patient be sarcopenic? Cite the following paper to support this discussion: DOI: 10.3390/nu17020282
  4. The authors should include a section on “De-frailing” interventions, covering strategies such as nutrition, exercise, rehabilitation, and hormonal replacement therapy. Cite DOI: 10.3390/diagnostics14192153 for general interventions, DOI: 10.3390/nu17020282 for sarcopenia-specific interventions, and DOI: 10.1007/s42000-024-00587-2 for testosterone therapy in anti-aging.
  5. The authors should also incorporate notable molecular and pathophysiological mechanisms associated with aging and frailty. Cite the following paper for this discussion: DOI: 10.3390/jpm14090931.

Author Response

Reviewer nº1.

Great paper! It provides a good overview and multiple perspectives on frailty syndrome, which remains a vague, poorly understood, and not yet well-established concept. However, there are some remarks that the authors should address:

The authors should discuss the definition of frailty as an accumulation of deficits and provide insights into tools like the Rockwood Frailty Index and similar indices.

Response: Thank you for the comment. We have made modification sin section “3.1. Historical footsteps of frailty”, providing further insights into de distinction of the accumulation of deficits model from the phenotype approach to frailty (current lines 104-117).

They should elaborate on the associations of frailty with specific diseases. For example, frailty is observed in up to 50% or more of patients with heart failure (HF) but is less prevalent in other chronic diseases. This could offer valuable context.

Response: Thank you for the comment. The present review is focused on the construct of frailty into older adults (geriatrics), and the efforts and discussion of the present review was addressed to this target population, as it was the original population where it was originated. In any case, we have made reference to the extension of the construct of frailty to other clinical population such as breast cancer or heart failure (current lines 118-124).

The manuscript should delve deeper into the relationship between frailty and sarcopenia, including underlying mechanisms. Why is frailty distinct from sarcopenia? Can a patient be frail but not sarcopenic? Conversely, can a robust patient be sarcopenic? Cite the following paper to support this discussion: DOI: 10.3390/nu17020282

Response: Thank you for your valuable feedback. Our review is structured as a critical analysis of the concept of frailty and we have only discussed its overlap with other constructs such as morbidity, function or disability, as stated in the sections of the manuscript. We have considered the discussion of these constructs because of its iteration in literature and the relevant overlap between them. Following your recommendation in the following comments, and the ones from Reviewer nº2, we have added a new section of the etiology of frailty, making allusions to how the concept of sarcopenia may derive (as an anthropometric, and functional measure) to the development of frailty, along with the presence of several immune and endocrine biomarkers (current lines 775-789)

The authors should include a section on “De-frailing” interventions, covering strategies such as nutrition, exercise, rehabilitation, and hormonal replacement therapy. Cite DOI: 10.3390/diagnostics14192153 for general interventions, DOI: 10.3390/nu17020282 for sarcopenia-specific interventions, and DOI: 10.1007/s42000-024-00587-2  for testosterone therapy in anti-aging.

Response: Thank you for your thoughtful suggestion. Our manuscript is structured as a critical review of the concept of frailty, focusing exclusively on its definitions, assessment instruments, and its relationship with other constructs based on existing definitions. The primary aim of this review is to critically analyze how frailty has been conceptualized over time, emphasizing the challenges and inconsistencies in its measurement.

Including a dedicated section on “de-frailing” interventions could divert attention from our primary objective, which is to provide a critical analysis of frailty as a construct rather than exploring specific treatment strategies. We hope that the scope and purpose of our manuscript have been clearly conveyed, as our aim is not to expand into other research areas, such as frailty interventions, which fall outside the intended focus of this review.

The authors should also incorporate notable molecular and pathophysiological mechanisms associated with aging and frailty. Cite the following paper for this discussion: DOI: 10.3390/jpm14090931

Response: Thank you for the comment. This point has been addressed in section “3.7. Not a clinical entity nor a unique etiology”, discussing the etiology, pathophysiology along with several molecular basis on the development of frailty (current lines 775-789).

Reviewer 2 Report

Comments and Suggestions for Authors

Thank you for the opportunity to read this work, which provides a significant contribution to the field. It highlights the issues and concerns with defining and measuring frailty. The review is an important topic, and the paper comprehensively addresses many ideas surrounding frailty. Although it offers more questions than answers, it brings attention to numerous problems with the assessment of frailty. I appreciate the inclusion of the figures to illustrate concepts. Starting with the comparison of theory, concept, and operations is helpful. The paper could use some improved organization, as indicated below. If you could review the headings and explanations of figures, it would be helpful and reduce the cognitive load. It would be beneficial to provide a paragraph outlining current definitions of frailty, including the concept of "physiologic reserve," before moving into comparisons and assessments.

Author Response

Reviewer nº2.

Thank you for the opportunity to read this work, which provides a significant contribution to the field. It highlights the issues and concerns with defining and measuring frailty. The review is an important topic, and the paper comprehensively addresses many ideas surrounding frailty. Although it offers more questions than answers, it brings attention to numerous problems with the assessment of frailty. I appreciate the inclusion of the figures to illustrate concepts. Starting with the comparison of theory, concept, and operations is helpful. The paper could use some improved organization, as indicated below. If you could review the headings and explanations of figures, it would be helpful and reduce the cognitive load.

Response: Thank you for the suggestion. Changes have been conducted along the headings of the “Results” sections in order to facilitate the comprehension of our arguments. Changes have been conducted in the following sections:

  • Section 3.1. "Tracing the historical footsteps of frailty" was modified to "Historical footsteps on frailty", to provide a clear statement.
  • The subsequent sections were we addressed 3.2. "Bridging the gap between foundations and their materialization", 3.3. "Do we have theoretical, conceptual, and operational definitions of frailty?", and 3.4. "How the literature conceptualizes physical and multidimensional frailty?", were now modified as follows.
    • These 3 sections now belong to a broader section named as 3.2. “The construct of frailty”. This section starts explaining how to bridge the gap between foundations and materialization (but now, the title section is absent), to serve as an introduction for the following sections.
    • We have changed "Do we have theoretical, conceptual, and operational definitions of frailty?" for a straightforward statement as 3.2.1."Theoretical, Conceptual, and Operational definitions of frailty".
    • Section "How the literature conceptualizes physical and multidimensional frailty?" was modified to "Physical vs. multidimensional frailty"
  • Section 3.5. "Are current instruments valid for assessing frailty?" was modified to "Validity and limitations of frailty assessment tools" being more precise and neutral.
  • No changes were conducted on 3.5.1. Inconsistencies in instruments’ content, nor in 3.5.2. Inconsistencies in assessment procedures, scoring systems and weightings. Here we have additionally reordered the section sincluding in section 3.5 the “Heterogeneity in sampling profiles and in scale responsiveness”
  • Section “A misleading diagnosis overlapping with other constructs” was modified to “3.6. Frailty overlaps with many other constructs”, being more clear.
  • Section “Is frailty redundant with function, disability, or morbidity?” changes to “Frailty vs. disability and functioning”.
  • Section “Is really frailty a clinical entity?” changes to “Not a clinical entity nor a unique etiology”.
  • Section “Does frailty fit within Modern Medicine Paradigms?” changes to “Frailty in the context of Modern Medicine Paradigms”.
  • Section “Is frailty able to detect future health-related adverse outcomes?” changes to “Assessing the ability of frailty to predict adverse events”.

Thank you for the comment. Changes have been conducted in the foot-notes of the figures, further explaining what is being illustrated. These changes were conducted in Figure 1, Figure 2.

It would be beneficial to provide a paragraph outlining current definitions of frailty, including the concept of "physiologic reserve," before moving into comparisons and assessments.

Response: Thank you for your comment. We have included emerging concepts in geriatric care, such as physiologic reserve, intrinsic capacity, physical resilience, and functional ability, to provide readers with additional literature and context on the evolving understanding of frailty (current lines 125-130). Our goal is to present the historical development of frailty and how related terms have emerged over time, contributing to this field. However, we have not focused on a detailed discussion of these concepts to maintain our primary focus on analyzing and discussing frailty.

Please review the numbering of the pages and headers. Page 22 starts over with numbering.

Response: Thank you for the comment. The numbering has been corrected.

Reconsider the headings- especially the 3.xx sections. Some are questions, and some are not. It would be helpful if they were clear to show the content and less creative.

Response: Thank you for the suggestions. Changes have been conducted for headings as mentioned previously.

The titles are not parallel at each level, making it hard to follow your arguments at times. It is very important that your headings follow an organized format and clearly indicate the argument being made to help the reader follow you.

Response: Thank you for the suggestions. Changes have been conducted on the numbering of the headings, its content, and the position of sections along the manuscript as mentioned earlier.

The figures could benefit from titles that are more descriptive. Consider improving the figure titles and giving a brief description of their organization.

Response: Thank you for the comment. Changes have been made in all figures including a brief description of its content.

Lines 60-73. Your purpose and objectives overlap. Suggest making them more specific and reducing repetition.

Response: Thank you for the suggestion. We have eliminated the fist paragraph and maintained the 1.1. Objective section, as both were reiterative. In addition, several changes have been included in the objective section to improve the contextualization of the article.

109-112- repetitive- recommend delete

Response: Thank you for the comment. We have performed modifications in those lines (current lines 104-124), to reduce the repetitive message. However, we decided to maintain the message as that section alludes to a very relevant point, that concerns the original division between the physical and multidimensional frailty paradigms, which is not mentioned in other sections of the manuscript.

113-119- combine paragraphs into one

Response: Thank you for the comment. Those sections have been combined in one paragraph in current lines (current lines 118-124).

120: What exactly is frailty? It does not answer this question but moves immediately to 3.2.1 bridging the gap between foundations…. Providing a paragraph that discusses current definitions of frailty would be beneficial. You missed the opportunity to specifically discuss it as a loss of physiologic reserve in multiple systems. It would be helpful to revisit “loss of physiologic reserve” in the discussion about the difference between frailty and disability.

Response: Thank you for the comment. We have considered your point, changing the head title of the section to “3.2. The construct of frailty” starting with an introductory section of how constructs are materialized. It is important to mention, that the aim of this review was not to discuss possible definitions of frailty, as it has been already done in previous reviews (De Lekan et al., 2021; Sobhani et al., 2021), being mentioned in section “3.2.1. Theoretical, conceptual, and operational definitions of frailty”. We aimed to discuss the presence of types of definitions, as one of the gaps identified is the lack of a fundamental and unified theoretical definition, further concerning the heterogeneity in conceptual and operational definitions.

In any case, we have considered your point, generating a final section of Results, titled “3.11. A summary of findings with the proposal of a new definition”. Here we summarize the findings from previous definitions (mainly from De Lekan et al., 2021; Sobhani et al., 2021), and the findings from our content analysis, highlighting the strengths and weaknesses of the proposed definition, which tries to explain the current state of the frailty construct. We finally conclude with a definition of frailty.

156 extra space

Response: Thank you for the comment. The extra space was replaced by a single space (current lines 172).

159 “2” should be “two”

Response: Thank you for the comment. The proposed change has been conducted (current lines 175).

158-163 – please clarify. I am not clear on what you are stating. You indicate that 2 reviews extracted and synthesized the definitions, but the paragraph is focused on exploration of definitions and they types. You are losing me on the extracted and synthesized statement.

Response: Thank you for the comment. Modifications have been made in this section to clarify the message (current lines 174-177).

177- 199 could this be displayed more clearly? Perhaps a table with 3 columns that show similar concepts in rows and move to an appendix? It is hard to read in this format and distracts from the text. I believe Figure 2 is sufficient.

Response: Thank you for the suggestion. The original bulletpoints with every of the concepts written were eliminated maintaining exclusively the Figure 2.

208: is not a question

Response: Thank you for the comment. The heading was modified from “How the literature conceptualizes physical and multidimensional frailty?” to “Physical vs. multidimensional frailty”.

268: based on analysis of which frailty instrument content? Clarify the scope of inclusion. Explain the purpose and organization of Figure 4 to orient the reader. Figure 4 is unreadable at its current location. Recommend including as a full page landscape so information is readable

Response: Thank you for the comment. The Figure 4 has been resized to a larger scale in order to be readable. In addition, the text concerning section 3.5.1. Inconsistencies in instruments’ content has been rewritten and reorganized to further explain the process conducted.

272- please provide more information about Figure 5 and the purpose of including it. It would be helpful to orient the reader to its organization.

Response: Thank you for the comment. Changes were conducted in order to explain the information in Figure 5. These modifications were conducted in the Methodology section in order to provide the reader a context for what was conducted (see current line 87), and futher describe de findings described from the figure (see current lines 274-305).

275 typo

Response: Thank you for the comment. We have rewritten the footnote of Figure 5 correcting the typography.

288-290 – does not make sense

Response: Thank you for the comment. We have rewritten that section to be more comprehensible (current lines 325-326).

390-400- consider making this a chart for easier reading

Response: Thank you for the suggestion. Line charts have been included in the mentioned section to clarify the reading of each component defined in the ICF with another type of bullet points, while maintaining the same content (current lines 477-491).

444-469 organization could be improved. The bullets and C.1) format doesn’t follow previous formats and is confusing. Other locations where you number and bullet items is confusing.

Response: Thank you for the comment. Changes have been conducted in the labelling in this section, only including bullet points (current lines 534, 541, 548, 552, 559, 564).

3.4.2

1: consider changing this header (confusing)

Response: Thank you for your suggestion. Our intention was to make a comedic allusion to the well-known cinematic phrase "One ring to rule them all." However, we understand that the way it was written may have caused confusion. To ensure clarity, we have revised the title to "one construct to predict them all," which maintains the reference while being more straightforward.

15-18 language is too casual

Response: Thank you for the comment. Changes have been conducted in this section rewriting it with a more formal registry (current lines 692-706).

47-56: This paragraph is unclear to me. Remove items from the scale that predict the event is stated but in the last sentence it seems to contradict this point indicating that specific factors should be associated with the adverse factors. Loneliness and cognitive deficits do increase risk of falls and hip fractures. Please clarify this recommendation and maybe explain how it applies to Figure 9.

Response: Thank you for the suggestions. Changes along this section have been made to clarify our arguments (current lines 725-726, and line 731).

4.3 line 93: this continues to be unclear and does not seem feasible based on your other recommendations. Considering frailty as a loss of physiologic reserve across multiple systems requires that associated predicting events be considered.

Response: Thank you for the suggestion. An example has been provided to clarify the point made (current lines 945-947).

Page 3 of 38 (page 21) chart does not seem necessary

Response: Than you for the comment. We consider it relevant to provide that those risks were not assessed for combined frailty instruments (as Table 1 provides summary information of the predictive risk of combined frailty instruments despite their characteristics), and it does in Table 2. Therefore, readers and researchers can notice this gap in the literature and consider that they only have available this information for separate types of frailty instruments (Table 2).

Additional changes conducted:

Part of the content of section “3.6. Frailty overlaps with many other constructs” was moved to section “3.5.3 Heterogeneity in sampling profiles and in scale responsiveness”, to provide a more coherent argument in the present section concerning the heterogeneity of the samples identified.

In addition, modifications were conducted in section “3.6. Frailty overlaps with many other constructs” to further synthesise how frailty overlaps with other constructs reducing the redundancy with the previous sections.

We have additionally developed a graphical abstract showing the key findings from the article.

Round 2

Reviewer 1 Report

Comments and Suggestions for Authors

The authors did not modify the manuscript to respond to my comments 2-5.

Although I understand the scope of the paper, my previous comment 2 is necessary to give a practical sense and clinical context of frailty in common diseases. Please add one small paragraph.

In lines 775-789 there is no reference to sarcopenia (please add 1 paragraph to discuss the points I have suggested on my previous comment 3). This is very important.

In my opinion a small section for anti-fraiiling interventions is necessary. Any discussion of frailty is pointless without refering to potential interventions. I didnt ask the authors to change the scope of the paper, rather add a small ssction of 3-4 paragraphs for interventions. Please see the papers I have proposed in my previous comment 4 and refer shortly at least to nutrition, exercise, physiotherapy, and hormonal therapies. Here, again, is where the overlap between sarcopenia and frailty is crucial.

Author Response

The authors did not modify the manuscript to respond to my comments 2-5.

Although I understand the scope of the paper, my previous comment 2 is necessary to give a practical sense and clinical context of frailty in common diseases. Please add one small paragraph.

Response: Thank you for the comment. We have addressed the point in current lines 121-126, alluding to the extension of the frailty construct to clinical populations. We additionally mention that this process was extended from the adaptation of classical frailty tools from geriatics along with the development of targeted instruments for clinical populations, including patients with hip fracture, heart failure, breast cancer or stroke among others. We now can provide the reader a broader scope of the concept of frailty not only affecting older adults but specific clinical populations.

In lines 775-789 there is no reference to sarcopenia (please add 1 paragraph to discuss the points I have suggested on my previous comment 3). This is very important.

Response: Thank you for your comment. To provide a thorough discussion while maintaining the manuscript’s objectives, we have added a new section, “3.6.4. Frailty vs. Sarcopenia.” This section compares both constructs by examining their definitions, tracing the evolution of sarcopenia’s conceptualization, and highlighting changes in its diagnostic criteria over time. Additionally, we discuss how sarcopenia and frailty share common characteristics while remaining distinct entities. We have addressed this topic as it is a subject of ongoing debate in the literature and have incorporated the reviewer's request to discuss sarcopenia within the defined scope of the manuscript.

In my opinion a small section for anti-fraiiling interventions is necessary. Any discussion of frailty is pointless without refering to potential interventions. I didnt ask the authors to change the scope of the paper, rather add a small ssction of 3-4 paragraphs for interventions. Please see the papers I have proposed in my previous comment 4 and refer shortly at least to nutrition, exercise, physiotherapy, and hormonal therapies. Here, again, is where the overlap between sarcopenia and frailty is crucial.

Response: Thank you for your valuable feedback. We have generated a new section of “3.9.2. Utility for guiding specific interventions”, of 18 lines (current lines 819-837), where we expose the current treatments for managing frailty, and how they should be addressed in the modern medicine paradigms. We have discussed the effectiveness of therapies, such as exercise, following the scope of the manuscript (heterogeneity of samples identified). We have also alluded to therapies such as nutritional, pharmacological or multimodal interventions.

This has enhanced the section “3.9. Frailty in the context of Modern Medicine Paradigms“ as in order to include the suggestion of a section of therapies, we have now redivided the structure of this section into “3.9.1. Utility as a screening process not for a precise evaluation”, and “3.9.2. Utility for guiding specific interventions”.